# LOCALITY-AWARE PARALLEL DECODING FOR EFFICIENT AUTOREGRESSIVE IMAGE GENERATION

**Zhuoyang Zhang**[*]  **Luke J. Huang**[*]  **Chengyue Wu**  **Shang Yang**  **Kelly Peng**
**Yao Lu**  **Song Han**
MIT    NVIDIA    First Intelligence
https://github.com/mit-han-lab/lpd

## ABSTRACT

We present *Locality-aware Parallel Decoding* (LPD) to accelerate autoregressive image generation. Traditional autoregressive image generation relies on next-patch prediction, a memory-bound process that leads to high latency. Existing works have tried to parallelize next-patch prediction by shifting to multi-patch prediction to accelerate the process, but only achieved limited parallelization. To achieve high parallelization while maintaining generation quality, we introduce two key techniques: (1) **Flexible Parallelized Autoregressive Modeling**, a novel architecture that enables arbitrary generation ordering and degrees of parallelization. It uses learnable position query tokens to guide generation at target positions while ensuring mutual visibility among concurrently generated tokens for consistent parallel decoding. (2) **Locality-aware Generation Ordering**, a novel schedule that forms groups to minimize intra-group dependencies and maximize contextual support, enhancing generation quality. With these designs, we reduce the generation steps from 256 to 20 (256×256 res.) and 1024 to 48 (512×512 res.) without compromising quality on the ImageNet class-conditional generation, and achieving at least 3.4× lower latency than previous parallelized autoregressive models.

## 1 INTRODUCTION

Autoregressive modeling has achieved state-of-the-art results in large language models in terms of scalability and generalizability (Brown et al., 2020; OpenAI, 2023; Touvron et al., 2023a;b; Grattafiori et al., 2024; Jiang et al., 2024; Yang et al., 2024; 2025; Liu et al., 2024a).

Naturally, many works have applied this power-ful paradigm to visual generation (Esser et al., 2021; Lee et al., 2022; Ramesh et al., 2021; Yu et al., 2022; Sun et al., 2024; Tian et al., 2024). Moreover, this autoregressive formulation of vi-sual generation has become increasingly cru-cial for unified multimodal generation (OpenAI, 2025; Wang et al., 2024a; Wu et al., 2024c;a; Chen et al., 2025a; Ma et al., 2025; Jiao et al., 2025; Song et al., 2025; Chen et al., 2025b; Zhao et al., 2025; Lin et al., 2025; Deng et al., 2025; Liao et al., 2025; Xie et al., 2025) since it is highly compatible with language modeling.

Prevailing autoregressive visual generation methods typically follow two paradigms: (1) next-patch prediction by flattening the image into a sequence of patches (Esser et al., 2021) and (2) next-scale prediction via coarse-to-fine multi-scale representations (Tian et al., 2024).

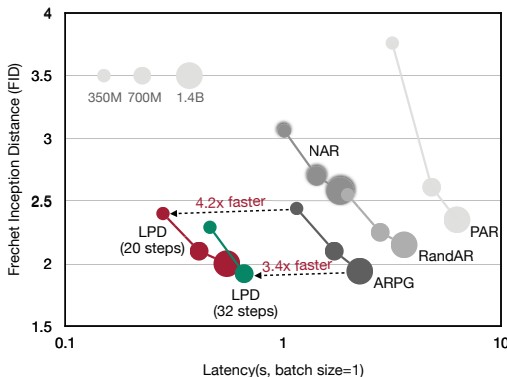

Figure 1: **Performance comparison among par-allelized autoregressive models on ImageNet 256×256.** We significantly reduce the generation steps and achieve at least 3.4x lower latency com-pared with previous models.

---

[*]Equal Contribution.

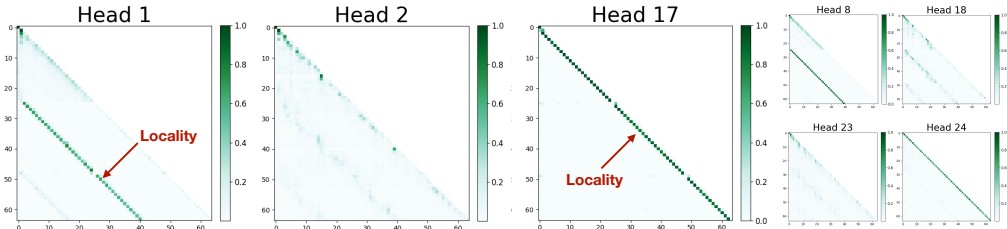

Figure 2: **Visualization of attention maps in the LLAMAGEN-1.4B model.** There is strong spatial locality, as the attention of a decoding token is concentrated on nearby spatial tokens. LLAMAGEN encodes images into $24 \times 24$ tokens, where a token that is 24 positions earlier in the attention map corresponds to the token directly above it in the 2D grid.

In the first formulation, generating one token per step creates a memory-bound workload[1], causing latency to scale with the number of steps. The second formulation substantially reduces generation steps and thus latency. However, its multi-scale token representation fundamentally differs from the universal flat token representation, making it incompatible with widely used flat vision perception foundation models (e.g., CLIP (Radford et al., 2021; Zhai et al., 2023), DINO (Caron et al., 2021; Oquab et al., 2023)) and thereby limiting interoperability with perception backbones that have been proven critical for unified multimodal systems (Wu et al., 2024c; Ma et al., 2025; Jiao et al., 2025; Song et al., 2025; Chen et al., 2025b; Zhao et al., 2025; Lin et al., 2025; Tong et al., 2024; Wu et al., 2025; 2024b).

Thus, autoregressive visual generation should be (1) highly efficient: minimizing latency and maximizing throughput; (2) remain flat token representations for universality and compatibility with vision backbones and, by extension, unified multimodal models. Recent works (Wang et al., 2024b; Pang et al., 2024; Li et al., 2025a) have tried to parallelize next-patch prediction by shifting to multi-patch prediction to accelerate the process, but only achieved limited parallelization. Non-autoregressive mask-prediction models like MASKGIT (Chang et al., 2022) enable multi-patch prediction but require full attention for bidirectional context, making them less efficient than autoregressive methods.

To address the challenges, we introduce *Locality-aware Parallel Decoding* (LPD), a framework that consists of a novel flexible parallelized autoregressive modeling architecture and a novel locality-aware generation order schedule. We design a new modeling architecture as conventional decoder-only autoregressive models struggle with flexible generation order and parallelization, limiting efficiency. In contrast, ours enables arbitrary generation order and degrees of parallelization. This is achieved by using learnable position query tokens to guide the model in generating tokens at target positions. Moreover, the generation is parallel-aware, as we leverage specialized attention mechanism to ensure mutual visibility among tokens generated concurrently. Notably, our design also inherits the KV caching mechanism, avoiding redundant computation.

Furthermore, we observe strong spatial locality in image generation attention where tokens predominantly attend to nearby regions as shown in Figure 2. This indicates a high dependency among nearby tokens, meaning that spatially closer tokens provide stronger conditioning. Recent works (Wang et al., 2024b; Besnier et al., 2025) also identify that minimizing mutual dependency among simultaneously generated tokens is essential to maintain sample consistency. With these insights, we introduce a locality-aware generation order schedule that selects parallel decoding groups to maximize contextual support while minimizing intra-group dependencies, enabling higher degrees of parallelization.

We examine the effectiveness of our proposed method on ImageNet class-conditional image generation. Our results reveal that we reduce the generation steps of traditional raster-order autoregressive generation from 256 to 20 (256×256 res.) and 1024 to 48 (512×512 res.) without compromising quality, and achieving at least 3.4× lower latency (Figure 1) than previous parallelized autoregressive models. Thanks to the design of flexible autoregressive modeling, our models are also capable of zero-shot image editing including class-conditional editing, inpainting and outpainting.

---

[1]A memory-bound workload refers to the scenario where the efficiency is limited by memory access speed rather than computation speed. In this context, each generation step requires loading the entire model parameters into GPU registers, making the process bottlenecked by memory bandwidth rather than computational power.

Figure 3: **Raster Order *vs.* Flexible Parallelized Autoregressive Modeling.** (a) In raster order, each token simultaneously provides context and predicts the next token, restricting flexibility and efficiency. (b) Our approach decouples these roles: previously generated tokens supply context, while position query tokens drive parallel generation at arbitrary target positions. This separation enables both flexible order and efficient parallelization.

## 2 METHOD

### 2.1 RETHINKING AUTOREGRESSIVE MODELING

In next-patch autoregressive modeling, images are split into patches and usually discretized via a tokenizer into image tokens. While the joint distribution of the $N$ tokens $x_1, \cdots, x_N$ and condition $c$ is extremely high dimensional and therefore hard to model directly, the autoregressive framework makes this amenable by factorizing the total joint distribution as

$$p(x_1, x_2, \ldots, x_N; c) = \prod_{n=1}^{N} p(x_n | x_{<n}; c) \tag{1}$$

The training objective of the autoregressive model is therefore to optimize parametric approximations $p_\theta(x_n | x_{<n}; c)$ for those one-step conditionals. This factorization needs a predefined order, typically raster order, as shown in Figure 3 (a). However, during sampling, this leads to $N$ sequential steps, creating a major efficiency bottleneck.

To reduce the number of sequential generation steps, we can partition tokens into $G$ disjoint groups $\{X_1, \cdots, X_G\}$, where each group $X_g = \{x_{g_1}, \cdots, x_{g_m}\}$ is predicted jointly, resulting in the following:

$$p(x_1, x_2, \ldots, x_N; c) = \prod_{g=1}^{G} p(X_g \mid X_{<g}; c) \tag{2}$$

The training objective becomes optimizing $p_\theta(X_g \mid X_{<g}; c)$. Previous work has shown that directly grouping tokens in raster order causes significant performance degradation (Wang et al., 2024b; Pang et al., 2024). This is because spatially adjacent tokens exhibit strong mutual dependencies, and independent sampling usually leads to generation inconsistencies inside a group. It is essential to break the raster order when grouping. In addition, the size of the prediction group $|X_g|$ should gradually increase. As the context size $|X_{<g}|$ grows, it offers stronger conditioning, allowing more tokens to be predicted in parallel. Previous work using masked transformers (Chang et al., 2022) also mirrors this intuition by predicting fewer tokens early when context is sparse and predicting more tokens over time. Therefore, an effective parallelized autoregressive model should support: (1) **Flexible generation order** to alleviate the issue caused by mutual interdependency of concurrently predicted tokens and (2) **Dynamic group sizes** increasing the number of tokens predicted per step with available context.

However, it is difficult to achieve these within the standard decoder-only autoregressive models, which are inherently designed with a fixed input-output structure, e.g. next-token prediction. In this modeling, each token simultaneously serves two roles: it provides **context** via its hidden state and enables **generation** via its output logits. This coupling limits flexibility in the the generation order and output size. To address these challenges, we propose a novel flexible parallelized autoregressive modeling which is able to support arbitrary generation order and degrees of parallelization.

### 2.2 FLEXIBLE PARALLELIZED AUTOREGRESSIVE MODELING

Our core idea is to decouple the context representation and token generation by leveraging separate tokens. We illustrate this in Figure 3 (b). In this formulation, previously generated tokens are encoded

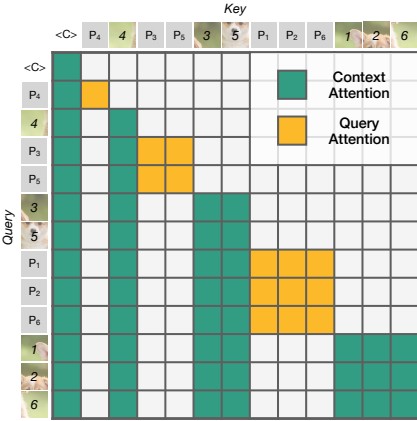

Figure 4: **Illustration of the training attention mask.** *Context Attention* allows subsequent tokens to attend to the context tokens causally. *Query Attention* ensures mutual visibility among the position query tokens within the same step, and prevents any subsequent tokens from attending to the query tokens. For example, image token 4 can be attended to by all subsequent tokens, including image tokens and position query tokens, to provide context information. The two position query tokens $P_3$ and $P_5$ in the same generation step attend to the condition, to the image token 4, and to each other, while ignoring the earlier query $P_4$.

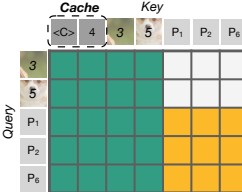

Figure 5: **Illustration of the inference attention mask.** *Encoding* with image tokens and *Decoding* with position query tokens can be fused into a single step. Taking step 2 in Figure 3 (b) as the example, it simultaneously encodes the previously generated image tokens 3, 5 to update the KV-cache and decodes the desired image tokens 1, 2 and 6 in parallel.

to provide context and the generation is driven by learnable position query tokens corresponding to the desired target positions. These position query tokens are constructed by adding the positional embedding of the target location to a shared learnable embedding. By directly inputting these position-specific queries, the model can generate tokens at arbitrary target positions in parallel. This design allows the model to leverage positional information in both the context and generation pathways, enabling arbitrary generation order.

**Training formulation.** We train the model to transform each position query token into the corresponding ground-truth image token, conditioned on all ground-truth tokens that precede it. To preserve teacher-forcing while allowing parallel prediction, we interleave position query tokens with ground-truth tokens and apply a specialized training attention mask as shown in Figure 4 that contains two attention patterns:

1. **Context Attention** allows subsequent tokens to attend to context tokens causally.
2. **Query Attention** ensures mutual visibility among the position query tokens within the same step, and prevents any subsequent tokens from attending to the query tokens.

**Inference formulation.** At test time we alternate between encoding the generated image tokens and decoding with position query tokens.

1. **Encoding.** Sampled image tokens go through a forward pass to store the KV cache, providing context for future decoding steps.
2. **Decoding.** Learnable position query tokens attend to all previously generated tokens in the KV cache, and the forward pass outputs logits for each target position in parallel. KV cache for query tokens is not stored.

However, sequentially execute these two operations double the generation steps. As shown in Figure 3 (b), these two operations can be fused into a single step via a specialized inference attention mask as shown in Figure 5.

**Comparison with other methods.** Recent efforts have also pursued parallel generation in autoregressive modeling, yet each carries inherent limitations. One line of work, exemplified by SAR (Liu et al., 2024b) and ARPG (Li et al., 2025a), adopts an encoder-decoder architecture where target-aware query tokens attend to the encoder's key-value cache via cross-attention. However, as illustrated in Figure 6 (a), the target positions themselves do not contribute any key-value pairs, resulting in the tokens generated within the same parallel step being produced independently of one another.

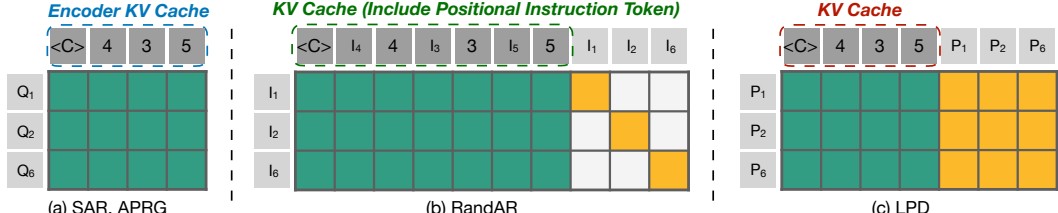

Figure 6: **Comparison with other methods.** (a) Encoder–decoder approaches such as SAR and ARPG generate tokens independently, since query tokens contribute no key–value pairs. (b) Decoder-only methods like RANDAR rely on positional instruction tokens, but the causal mask reduces parallel generation to batched next-token prediction and forces instruction tokens to be cached, doubling memory. (c) In contrast, our method employs a specialized training mask that ensures mutual visibility among concurrently predicted tokens while caching only the generated tokens.

Another approach, represented by RANDAR (Pang et al., 2024), adheres to the prevailing decoder-only architecture. It achieves arbitrary order by inserting positional instruction tokens to designate target positions. However, it still leverages a standard causal mask during training. This strategy, as depicted in Figure 6 (b), leads to two notable issues: (1) the parallel generation degenerates into a batched next-token prediction instead of joint prediction and (2) the positional instruction tokens must be stored in the KV cache during inference, doubling the memory consumption. Compared with these two methods, our method as shown in Figure 6 (c) guarantees the visibility among all concurrently predicted target positions and only stores the generated tokens in the KV cache.

PAR (Wang et al., 2024b), NAR (He et al., 2025), and ZipAR (He et al., 2024) preserve the standard decoder-only architecture and increase the number of tokens generated per step. Although they guarantee mutual visibility among concurrently generated tokens, they rely on a fixed parallel generation order, which prevents them from supporting arbitrary generation orders. This limits the generation flexibility thus achieved limited parallelization and generation quality. ACDIT (Hu et al., 2024) shares similar attention scheme with us, yet it was used for evenly interpolating between autoregressive and diffusion modeling.

## 2.3 LOCALITY-AWARE GENERATION ORDER SCHEDULE

To fully leverage our flexible parallelized autoregressive modeling architecture, we introduce a locality-aware generation order schedule. This schedule is guided by two key principles (1) **High proximity to previously generated tokens**: target positions should be spatially close to existing context to ensure strong conditioning and (2) **Low proximity among concurrently generated tokens**: tokens predicted in the same parallel step should be spatially distant to reduce mutual dependency.

These principles are derived from a systematic analysis of the attention patterns in autoregressive image generation by the widely adopted LLAMAGEN (Sun et al., 2024) model. Using LLAMAGEN, we generate 50,000 images and collect attention scores at each decoding step. Qualitative attention patterns are shown in Figure 2, and quantitative results are presented in Figure 7. To quantify locality, we define the *Per-Token Attention* (PTA) to a neighborhood of radius $s$ [2] as:

$$PTA_s = \frac{1}{N} \sum_{i=1}^{N} \frac{\sum_j \text{Attention}(T_i, T_j) \cdot \mathbb{I}[d(T_i, T_j) = s]}{\sum_j \mathbb{I}[d(T_i, T_j) = s]} \tag{3}$$

where $\text{Attention}(T_i, T_j)$ denotes the attention weight from token $T_i$ to token $T_j$, and $d(T_i, T_j)$ is their Euclidean distance on the 2D image grid.

As shown in Figure 7 (a), PTA decreases sharply with increasing distance, indicating a strong spatial locality in the attention mechanism. This suggests that nearby tokens carry significantly more useful information during decoding, and that spatially adjacent tokens are highly dependent on one another for accurate prediction. This locality pattern is consistently observed across all attention heads. In Figure 7 (b), we visualize the *Attention Sum*, defined as the total attention score a decoding token assigns to tokens within a relative distance $s$. The plot uses $s = 3$ and confirms that most attention is concentrated within local neighborhoods, reinforcing the importance of spatial locality.

---

[2] The neighborhood is defined as the set of tokens whose centers are exactly a euclidean distance of $s$ away.

This analysis supports our two principles: decoding tokens should remain close to previously generated tokens to maximize contextual support, and distant from concurrently generated tokens to minimize intra-group dependency.

Based on these principles, we implement a locality-aware generation order schedule described in Algorithm 1. Suppose we use $K$ decoding steps to generate $N^2$ tokens, with group sizes $O = [o_1, o_2, \ldots, o_K]$, where $o_k$ is the number of tokens generated in step $k$, typically increasing via a cosine schedule. At each step $k$, we compute the euclidean distance between unselected and already selected tokens to measure spatial proximity, where closer distance leads to higher proximity. We sort unselected tokens by proximity and split them into two sets: $c_1$ are tokens with sufficient proximity larger than the threshold $\tau$ which are eligible for the following high-proximity selection, and $c_2$ are the rest. We sequentially select tokens from $c_1$, adding each to the selected set while filtering out nearby tokens that the relative distance is smaller than the repulsion threshold $\rho$, which are added to $c_2$. If all the grids in $c_1$ are considered and the number of selected grids is less than $o_k$, we use farthest point sampling (Qi et al., 2017) to select the remaining grids from $c_2$ to ensure spatial low dependency. It is worth noting that the generation order can be precomputed and stored for direct use during inference, incurring no additional latency. We provide the PyTorch implementation in Appendix B.1. We further clarify the key distinction between our method and prior work in Appendix B.2.

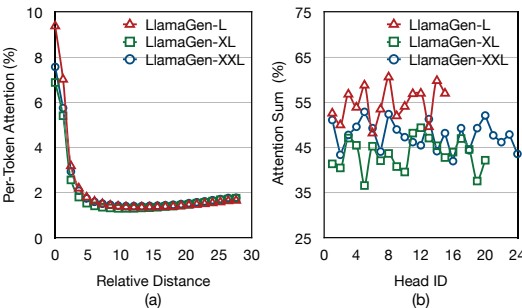

Figure 7: **Attention Analysis of LLAMAGEN.** (a) Attention diminishes with distance (b) Spatial locality is consistently observed in all heads.

---

**Algorithm 1:** Locality-aware Generation Order Schedule

**Input:** decoding steps $K$, group sizes $O = [o_1, o_2, \ldots, o_K]$, grids $G = \{(i,j)\}_{i,j=1}^{N}$, proximity threshold $\tau$, repulsion threshold $\rho$;

schedule $S = [\,]$;
**for** $k = 1, \ldots, K$ **do**
    $s = [\,]$;
    $p = 1/\operatorname{euclidean}(G \setminus S, S)$;             ▶ proximity measurement
    $c = \operatorname{sorted}(G \setminus S, key = p, reverse = True)$;
    $c_1, c_2 = \operatorname{cutoff}(c, \tau)$;
    **while** $\operatorname{len}(s) < o_k$ and $\operatorname{len}(c_1) > 0$ **do**
        $s = \operatorname{queue\_push}(s, \operatorname{queue\_pop}(c_1, 1))$;     ▶ high-proximity selection
        $c_1, f = \operatorname{filter}(c_1, s, \rho)$;
        $c_2 = \operatorname{queue\_push}(c_2, f)$;
    **if** $\operatorname{len}(s) < o_k$ **then**
        $s = \operatorname{queue\_push}(s, \operatorname{farthest\_point\_sampling}(c_2, s, o_k - \operatorname{len}(s)))$;
                        ▶ low-dependency selection
    $S = \operatorname{queue\_push}(S, s)$;
**return** $S$

---

For intuitive understanding, we illustrate an example of our generation order schedule in Figure 8. We also plot the schedule for raster order, random order and Halton order (Besnier et al., 2025) for comparison. The raster order generates tokens in a raster-scan manner and the random order generates tokens in a random manner. The Halton order is a low-discrepancy sequence to arrange the generation positions which spreads out the tokens to achieve uniform image coverage step by step.

## 3 EXPERIMENT

### 3.1 SETUP

**Models.** For fair comparisons with existing autoregressive image generation methods, we use the LLAMAGEN tokenizer (Sun et al., 2024) with codebook size 16384 and downsample factor 16. We train three models of different sizes: 337M, 752M, and 1.4B parameters. We use a standard

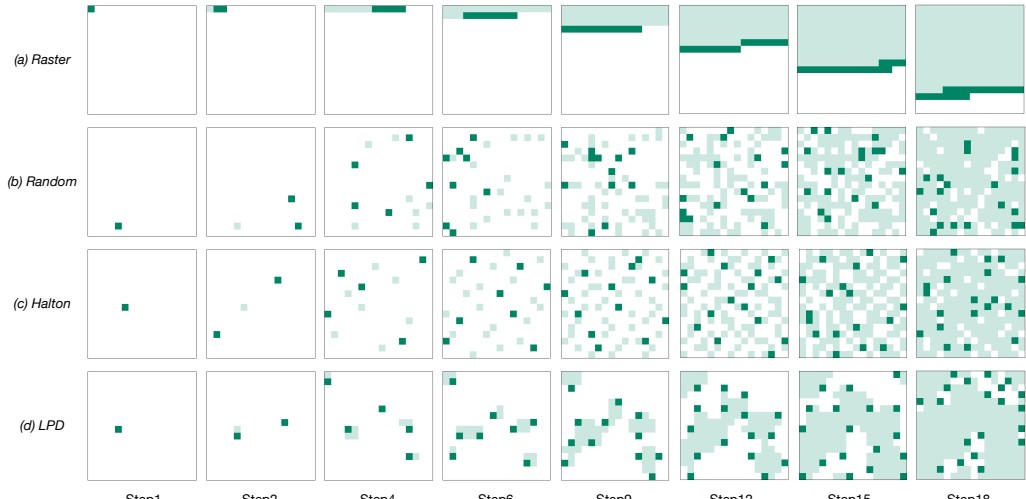

Figure 8: **Illustration of different generation order schedules.** All schedules leverage 20 decoding steps for $16^2$ tokens. Dark green marks newly selected grids and light green marks those already selected. Compared to others, our schedule selects grids close to previous ones and far from concurrent ones, maximizing the contextual support and minimizing the mutual dependency.

decoder-only transformer architecture, and refer to them as LPD-L, LPD-XL, and LPD-XXL, respectively. Please refer to the Appendix A.1 for more details.

**Training and Evaluation.** We train and evaluate our models on the class-conditional ImageNet (Russakovsky et al., 2015) 256×256 and ImageNet 512×512 datasets. We first train all models on ImageNet 256×256 for 450 epochs, with 50 epochs of learning rate warmup followed by constant learning rate and finally 50 epochs of cosine decay. For 512-resolution models, we load the pre-trained 256-resolution models and interpolate the positional embeddings and continue training on ImageNet 512×512 for another 50 epochs. During training, the image tokens are randomly shuffled while the class token is kept at the beginning. We train on a range of predefined decoding steps where the tokens per step follows a cosine schedule. We reportuse Fréchet Inception Distance (FID) (Heusel et al., 2017) as the primary metric computed on 50k,000 generated samples as the primary metric as well asnd also report Inception Score (IS) (Salimans et al., 2016), Precision, and Recall (Kynkäänniemi et al., 2019). Please refer to the Appendix A.2 for more details.

**Efficiency Profiling.** We profile all the efficiency results on a single NVIDIA A100 GPU with BFloat16 precision. We measure the latency with a batch size of 1 and throughput with a batch size of 64. We report the average latency over 500 inference steps, with a 100-step warm-up period.

## 3.2 MAIN RESULTS

We compare our models against a broad set of generative baselines on ImageNet 256×256 (Table 1). For a fair comparison, we also create a raster order counterpart following the same setup. As shown in the table, we reduce the generation steps from 256 to 20, achieving 12.8× generation steps reduction, without sacrificing the generation quality. Compared with other parallelized autoregressive models, we achieve significantly better image generation quality and efficiency. Taking LPD-XL model as an example, it achieves a FID of 2.10 with only 20 steps, reducing the number of generation steps by 3.2× compared to ARPG and achieving 4.2× lower latency. Increasing the steps slightly to 32 yields a FID of 1.92, even matching ARPG-XXL, while reducing latency by 3.4×. We further report our results on ImageNet 512×512 (Table 2). As shown in the table, we reduce the generation steps from 1024 to 48, achieving 21.3× generation steps reduction, without sacrificing the generation quality. These results validate the effectiveness of our flexible parallelized autoregressive modeling and the locality-aware generation order schedule. We also provide qualitative generation and zero-shot editing results in Figure 12.

Our method is general and can be readily extended to higher-resolution text-to-image generation (e.g., 1024×1024). We provide a detailed description of this extension in the Appendix A.3. We evaluate on the widely used GenEval (Ghosh et al., 2023) benchmark and report results in Table 3.

Table 1: **System-level comparison on ImageNet 256×256 class-conditional generation.** We evaluate the generation quality by metrics including Fréchet inception distance (FID), inception score (IS), precision and recall. #Steps is the number of model runs needed to generate an image. We measure latency with a batch size of 1 and throughput with a batch size of 64 on a single NVIDIA A100 GPU under BFloat16 precision, with classifier-free guidance (CFG) for both.

| Type | Model | #Para. | FID↓ | IS↑ | Precision↑ | Recall↑ | #Steps | Latency(s)↓ | Throughput(img/s)↑ |
|---|---|---|---|---|---|---|---|---|---|
| Diffusion | ADM-G [17] | 554M | 4.59 | 186.7 | 0.82 | 0.52 | 250 | – | – |
| | CDM [29] | – | 4.88 | 158.7 | – | – | 8100 | – | – |
| | LDM-4 [56] | 400M | 3.60 | 247.7 | – | – | 250 | – | – |
| | DiT-XL/2 [50] | 675M | 2.27 | 278.2 | 0.83 | 0.57 | 250 | 4.34 | 0.58 |
| | SiT-XL/2 [44] | 675M | 2.06 | 270.3 | 0.82 | 0.59 | 250 | – | – |
| Mask | MaskGIT [8] | 227M | 6.18 | 182.1 | 0.80 | 0.51 | 8 | – | – |
| | MAGVIT-v2 [83] | 307M | 1.78 | 319.4 | – | – | 64 | – | – |
| | MaskBit [72] | 305M | 1.62 | 338.7 | – | – | 64 | 1.03 | 5.39 |
| | MAR-B [37] | 208M | 2.31 | 281.7 | 0.82 | 0.57 | 64 | 18.14 | 2.93 |
| | MAR-L [37] | 479M | 1.78 | 296.0 | 0.81 | 0.60 | 64 | 20.80 | 2.11 |
| | MAR-H [37] | 943M | 1.55 | 303.7 | 0.81 | 0.62 | 64 | 25.96 | 1.45 |
| VAR | VAR-d16 [64] | 310M | 3.30 | 274.4 | 0.84 | 0.51 | 10 | 0.12 | 70.58 |
| | VAR-d20 [64] | 600M | 2.57 | 302.6 | 0.83 | 0.56 | 10 | 0.15 | 52.53 |
| | VAR-d24 [64] | 1.0B | 2.09 | 312.9 | 0.82 | 0.59 | 10 | 0.17 | 39.30 |
| | VAR-d30 [64] | 2.0B | 1.92 | 323.1 | 0.82 | 0.59 | 10 | 0.26 | 25.89 |
| AR | VQGAN-re [18] | 1.4B | 5.20 | 280.3 | – | – | 256 | – | – |
| | RQTran.-re [34] | 3.8B | 3.80 | 323.7 | – | – | 256 | – | – |
| | LlamaGen-L [61] | 343M | 3.07 | 256.1 | 0.83 | 0.52 | 576 | 12.22 | 2.08 |
| | LlamaGen-XL [61] | 775M | 2.62 | 244.1 | 0.80 | 0.57 | 576 | 18.51 | 1.14 |
| | LlamaGen-XXL [61] | 1.4B | 2.34 | 253.9 | 0.80 | 0.59 | 576 | 24.40 | 0.72 |
| | LlamaGen-3B [61] | 3.1B | 2.18 | 263.3 | 0.81 | 0.58 | 576 | 12.37 | 0.58 |
| | RAR-B [84] | 261M | 1.95 | 290.5 | 0.82 | 0.58 | 256 | 4.18 | 13.76 |
| | RAR-L [84] | 461M | 1.70 | 299.5 | 0.81 | 0.60 | 256 | 4.04 | 12.63 |
| | RAR-XL [84] | 955M | 1.50 | 306.9 | 0.80 | 0.62 | 256 | 5.47 | 8.76 |
| | RAR-XXL [84] | 1.5B | 1.48 | 326.0 | 0.80 | 0.63 | 256 | 6.59 | 6.72 |
| Parallelized AR | PAR-L-4× [71] | 343M | 3.76 | 218.9 | 0.84 | 0.50 | 147 | 3.16 | 6.83 |
| | PAR-XL-4× [71] | 775M | 2.61 | 259.2 | 0.82 | 0.56 | 147 | 4.79 | 3.69 |
| | PAR-XXL-4× [71] | 1.4B | 2.35 | 263.2 | 0.82 | 0.57 | 147 | 6.26 | 2.33 |
| | PAR-3B-4× [71] | 3.1B | 2.29 | 255.5 | 0.82 | 0.58 | 147 | 3.29 | 2.32 |
| | RandAR-L [48] | 343M | 2.55 | 288.8 | 0.81 | 0.58 | 88 | 1.97 | 28.59 |
| | RandAR-XL [48] | 775M | 2.25 | 317.8 | 0.80 | 0.60 | 88 | 2.78 | 17.06 |
| | RandAR-XXL [48] | 1.4B | 2.15 | 322.0 | 0.79 | 0.62 | 88 | 3.58 | 11.49 |
| | ARPG-L [36] | 320M | 2.44 | 291.7 | 0.82 | 0.55 | 32 | 0.58 | 104.92 |
| | ARPG-L [36] | 320M | 2.44 | 287.1 | 0.82 | 0.55 | 64 | 1.15 | 54.70 |
| | ARPG-XL [36] | 719M | 2.10 | 331.0 | 0.79 | 0.61 | 64 | 1.71 | 36.53 |
| | ARPG-XXL [36] | 1.3B | 1.94 | 339.7 | 0.81 | 0.59 | 64 | 2.24 | 26.23 |
| | NAR-L [27] | 372M | 3.06 | 263.9 | 0.81 | 0.53 | 31 | 1.01 | 41.03 |
| | NAR-XL [27] | 816M | 2.70 | 277.5 | 0.81 | 0.58 | 31 | 1.42 | 23.36 |
| | NAR-XXL [27] | 1.5B | 2.58 | 293.5 | 0.82 | 0.57 | 31 | 1.88 | 15.20 |
| AR | Raster Counterpart-L | 337M | 2.48 | 278.0 | 0.81 | 0.58 | 256 | 3.73 | 17.53 |
| | Raster Counterpart-XL | 752M | 2.12 | 307.4 | 0.81 | 0.60 | 256 | 5.29 | 12.31 |
| | Raster Counterpart-XXL | 1.4B | 2.01 | 316.0 | 0.80 | 0.59 | 256 | 7.10 | 8.99 |
| Parallelized AR | LPD-L | 337M | 2.40 | 284.5 | 0.81 | 0.57 | 20 | 0.28 | 139.11 |
| | LPD-XL | 752M | 2.10 | 326.7 | 0.80 | 0.59 | 20 | 0.41 | 75.20 |
| | LPD-XXL | 1.4B | 2.00 | 337.6 | 0.80 | 0.60 | 20 | 0.55 | 45.07 |
| | LPD-L | 337M | 2.29 | 282.7 | 0.81 | 0.58 | 32 | 0.46 | 110.34 |
| | LPD-XL | 752M | 1.92 | 319.4 | 0.79 | 0.61 | 32 | 0.66 | 61.24 |

As demonstrated in the table, LPD reduces the sampling steps for 1024×1024 image generation from 4096 to just 64 and simultaneously improves the GenEval score. This provides compelling evidence that LPD is a generalizable method capable of supporting high-resolution text-to-image generation. We also include qualitative generation results in Figure 13 in the appendix.

## 3.3 EFFICIENCY ANALYSIS

Our method introduces position query tokens to enable flexible generation. These tokens add extra queries and thereby increase FLOPs. However, the resulting computational overhead has a negligible impact on wall-clock latency in memory-bound settings such as small-batch inference. In these scenarios, the reduction in generation steps translates almost linearly into latency reduction. As the batch size increases, the system progressively shifts toward a compute-bound regime, where the additional overhead begins to matter and diminish the speedup. We provide a quantitative analysis in Figure 14 to illustrate this trend. By gradually increasing the batch size until reaching the memory limit, we observe that the model transitions from memory-bound to compute-bound when the batch size exceeds 16. Nevertheless, even at the maximum feasible batch size, our method retains a throughput advantage of approximately 3× over the raster-order baseline.

Table 2: **System-level comparison on ImageNet 512×512 class-conditional generation.** Metrics and evaluation setup are the same as in Table 1.

| Type | Model | #Para. | FID↓ | IS↑ | Precision↑ | Recall↑ | #Steps | Latency(s)↓ | Throughput(img/s)↑ |
|---|---|---|---|---|---|---|---|---|---|
| Diffusion | ADM-G [17] | 554M | 7.72 | 172.71 | 0.87 | 0.42 | 250 | - | - |
|  | DiT-XL/2 [50] | 675M | 3.04 | 240.82 | 0.84 | 0.54 | 250 | 11.32 | 0.10 |
|  | SiT-XL/2 [44] | 675M | 2.62 | 252.21 | 0.84 | 0.57 | 250 | – | – |
| Mask | MaskGIT [8] | 227M | 7.32 | 156.0 | 0.78 | 0.50 | 12 | – | – |
|  | MAGVIT-v2 [83] | 307M | 1.91 | 324.3 | - | - | 64 | – | – |
|  | MAR-L [37] | 481M | 1.73 | 279.9 | – | – | – | – | – |
| VAR | VAR-$d$36-s [64] | 2.3B | 2.63 | 303.2 | – | – | 10 | 0.45 | OOM |
| AR | VQGAN [18] | 227M | 26.52 | 66.8 | 0.73 | 0.31 | 1024 | – | – |
| Parallelized AR | ARPG-XL [36] | 719M | 3.38 | 257.8 | – | – | – | – | – |
| AR | Raster Counterpart-L | 337M | 2.54 | 278.5 | 0.80 | 0.58 | 1024 | 14.25 | 3.79 |
|  | Raster Counterpart-XL | 752M | 2.09 | 315.0 | 0.81 | 0.57 | 1024 | 20.93 | 2.36 |
| Parallelized AR | LPD-L | 337M | 2.54 | 292.2 | 0.81 | 0.55 | 48 | 0.69 | 35.16 |
|  | LPD-XL | 752M | 2.10 | 326.0 | 0.80 | 0.63 | 48 | 1.01 | 18.18 |

Table 3: **System-level comparison on GenEval 1024×1024 text-to-image generation.** Detailed per-category GenEval scores are provided in Table 7.

| Type | Model | #Para. | GenEval Score↑ | #Steps | Latency(s)↓ | Throughput(img/s)↑ |
|---|---|---|---|---|---|---|
| AR | Raster Counterpart-L | 344M | 0.55 | 4096 | 64.7 | 0.20 |
|  | Raster Counterpart-XL | 760M | 0.60 | 4096 | 93.8 | 0.14 |
| Parallelized AR | LPD-L | 344M | 0.58 | 64 | 1.01 | 5.28 |
|  | LPD-XL | 760M | 0.62 | 64 | 1.53 | 2.85 |

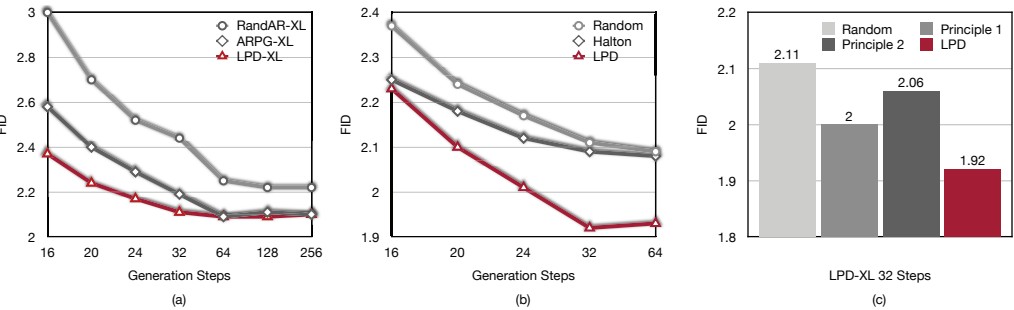

Figure 9: **Ablation Studies.** All ablation experiments are conducted with XL size models on 256×256 resolution. (a) Effectiveness of flexible parallelized autoregressive modeling. (b) Effectiveness of locality-aware generation order schedule. (c) Effectiveness of the locality principles.

## 4 ABLATION

**Effectiveness of Flexible Parallelized Autoregressive Modeling.** One key design of our flexible parallelized autoregressive modeling is the guarantee of the mutual visibility among all concurrently generated tokens. This is critical to maintain the consistency in the same group when the degree of the parallelization is high. We show the effectiveness of this design in Figure 9 (a). We compare our model with RANDAR and ARPG which lack this design. To only ablate the effectiveness of our flexible parallelized autoregressive modeling, we use random generation order for all models without our locality-aware parallel generation order schedule. As shown in the figure, with the generation steps decrease and the parallelization increases, our model exhibits a smaller FID increase compared with the other two models. For example, with 32 steps, our model almost maintain the performance with 256 steps but ARPG and RANDAR have a significant FID increase. This design is crucial for us to achieve fewer generation steps while maintaining the generation performance.

**Effectiveness of Locality-aware Generation Order Schedule.** We compare our schedule with another two generation order schedules as shown in Figure 9 (b). Random order just arrange the generation positions randomly. Halton order leverages the Halton low-discrepancy sequence to arrange the generation positions which spreads out the tokens to achieve uniform image coverage step by step. Intuitively it mainly focus on reducing the dependency inside a parallel group which shares the same insight with our second principle that low proximity is needed among concurrently generated tokens. However, the low-discrepancy sequence omits the importance of the already

generated context which is our first principle that we need to maintain high proximity to previously generated tokens. As shown in the figure, our locality-aware parallel decoding order consistently outperforms the other two orders, showing the effectiveness of our method.

**Effectiveness of the Locality Principles.** As introduced in Section 2.3, our locality-aware generation order schedule is guided by two principles. We ablate the effectiveness of these two principles in Figure 9 (c). As shown, the random order baseline yields an FID of 2.11. We first apply Principle 1 only, selecting points close to previously generated tokens without considering their mutual dependency. This improves the performance to 2.00. We then apply Principle 2 alone, using farthest point sampling at each step to ensure concurrently generated tokens are well separated, without considering context from previously generated tokens. This improves the FID to 2.06. Combining both in our locality-aware generation order achieves 1.92, highlighting the synergy of both principles. We further provide a sensitivity analysis of the hyperparameters $\tau$ and $\rho$ in Appendix B.3.

## 5 RELATED WORKS

### 5.1 AUTOREGRESSIVE IMAGE GENERATION

Autoregressive models generate the current output conditioned only on previous outputs. Usually this dependency is captured by causal attention mechanisms, enabling efficient inference via KV caching. Autoregressive modeling with GPT-style "next-token-prediction" (Brown et al., 2020; OpenAI, 2023; Touvron et al., 2023a;b; Chiang et al., 2023; Jiang et al., 2024) has dominated the field of language generation. Inspired by this success, autoregressive visual generation has shifted from operating on sequences of pixels (Van Den Oord et al., 2016; Van den Oord et al., 2016; Parmar et al., 2018; Chen et al., 2018; Salimans et al., 2017; Yu et al., 2021; Li et al., 2025b) to sequences of latent discrete tokens (Esser et al., 2021; Lee et al., 2022; Ramesh et al., 2021; Razavi et al., 2019; Yu et al., 2021; 2022; Sun et al., 2024; Yu et al., 2024; Wang et al., 2024a; Teng et al., 2024; Ren et al., 2025; He et al., 2025; 2024). However, the token-by-token decoding strategy is often bottlenecked by memory bandwidth. This limitation prevents full utilization of computation and results in high latency. Recently, "next-scale-prediction" (Tian et al., 2024; Han et al., 2024) has emerged to predict the next scale of the image instead of the next token thus accelerates the generation process. However, its multi-scale token representation fundamentally differs from the universal flat token representation, making it incompatible with widely used flat vision perception foundation models.

### 5.2 PARALLEL GENERATION IN SEQUENCE MODELING

Parallel generation has been widely studied in the field of language modeling. Prior to the era of large language models, masked-prediction architectures (Gu et al., 2017; Ghazvininejad et al., 2019; Gu et al., 2019) were commonly used to do parallel generation and iterative refinement. Recently, with the rapid success of large language models, speculative decoding (Chen et al., 2023; Leviathan et al., 2023) and its derivatives (Cai et al., 2024; Ankner et al., 2024) employ a draft model to generate the next few tokens and then the main model conducts the verification. In visual generation, masked-prediction models (Chang et al., 2022; Yu et al., 2023a;b; Chang et al., 2023) are widely used to generate masked tokens step by step leveraging a masked prediction transformer similar to BERT (Devlin et al., 2019; Bao et al., 2021; He et al., 2022), which are able to generate multiple tokens in parallel. However, they are non-autoregressive models and need bidirectional attention which is computationally expensive and KV cache is not applicable to accelerate the inference. Recent works (Wang et al., 2024b; Pang et al., 2024; Li et al., 2025a; He et al., 2025) have explored parallel generation in autoregressive models, but with limited parallelization and generation quality. Our proposed method enables greater parallelization without sacrificing performance.

## 6 CONCLUSION

Our contributions lie in two key aspects: (1) flexible parallelized autoregressive modeling and (2) locality-aware generation order schedule. We significantly reduce the generation steps required by the traditional autoregressive models without compromising the generation quality and achieve at least $3.4\times$ lower latency than previous parallelized autoregressive models.

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

# APPENDIX

## A  ADDITIONAL IMPLEMENTATION DETAILS

### A.1  MODEL ARCHITECTURE

We provide the model architecture configurations in Table 4. All models use a standard decoder-only transformer architecture. We vary model scale by adjusting the number of layers, the hidden size, and the number of attention heads.

| Model | Parameters | Layers | Hidden Size | Heads |
|-------|-----------|--------|-------------|-------|
| LPD-L | 111M | 12 | 1024 | 12 |
| LPD-XL | 775M | 36 | 1280 | 20 |
| LPD-XXL | 1.4B | 48 | 1536 | 48 |

Table 4: Model architecture configurations.

### A.2  TRAINING AND EVALUATION DETAILS

We train all models on ImageNet $256\times256$ for 450 epochs, with 50 epochs of learning rate warmup followed by constant learning rate and finally 50 epochs of cosine decay. For 512-resolution models, we load the pre-trained 256-resolution models and interpolate the positional embeddings and train on ImageNet $512\times512$ for 50 epochs. The continued training is conducted for 50 epochs using a cosine learning rate decay schedule, preceded by 1 epoch of warm-up. We use batch size 512 for LPD-L and 256 for LPD-XL.

We take the training of LPD-L model on $256 \times 256$ resolution as an example and list all the training hyper-parameters in Table 5. For LPD-XL and LPD-XXL, we use batch size 1024 and the same base learning rate.

| Hyper-parameters for $256\times256$ training | Configuration |
|---------------------------------------------|---------------|
| optimizer | AdamW |
| $\beta_1$ | 0.9 |
| $\beta_2$ | 0.95 |
| learning rate[3] | $8 \times 10^{-4}$ |
| batch size | 2048 ($64 \times 32$ GPUs) |
| training precision | BFloat16 |
| total epochs | 450 |
| warm-up epochs | 50 |
| constant LR epochs | 350 |
| cosine decay epochs | 50 |
| offsets | random per-sample |

Table 5: Training hyper-parameters for LPD-L on $256 \times 256$ resolution.

We train on a range of predefined decoding steps where the number of tokens in each step is determined by a cosine schedule. For the $256 \times 256$ resolution, the decoding steps are randomly selected from the set $\{8, 12, 16, 20, 24, 32, 64, 128, 256\}$. For the $512 \times 512$ resolution, the decoding steps are randomly selected from the set $\{32, 40, 48, 56, 64, 80, 96, 128, 160, 192, 224, 256, 512, 1024\}$. Take 20 steps in the $256 \times 256$ resolution as an example, the number of tokens in each step is [1, 2, 4, 5, 7, 8, 10, 11, 12, 14, 15, 16, 17, 18, 18, 19, 19, 20, 20, 20].

For evaluation, we sweep the optimal classifier-free guidance scale with an interval of 0.1 and follow the Locality-aware Generation Order Schedule.

### A.3  TEXT-TO-IMAGE GENERATION

We provide implemenation details in terms of model architecture, data, and training procedure.

**Model.** To process user prompts rather than class labels, we incorporate the Gemma-2-2B (Team et al., 2024) model as the text encoder, which converts prompts into text embeddings. These embeddings

---

[3]Effective LR computed as base lr $\times$ (global batch size$/256$) with base lr $= 1 \times 10^{-4}$.

are passed through a linear projection layer and then concatenated with the image embeddings, with the text embeddings placed at the beginning of the sequence. All other components remain identical to our class-conditioned models. Our locality-aware generation order schedule continues to apply, as only the spatial resolution needs to be increased.

**Data.** We use an internal MidJourney-style synthetic dataset containing approximately 10M images. All images are re-captioned using Qwen2.5-VL-32B (Bai et al., 2025), which produces both a concise caption and a detailed caption for each image. During training, we randomly sample one of the two captions to serve as the text prompt.

**Training Recipe.** We train two models, LPD-L (344M) and LPD-XL (760M), using a three-stage progressive-resolution schedule. We first train at 256×256 resolution for 40 epochs with batch sizes of 2048/1024 for the L/XL models. We then increase the resolution to 512×512 by loading the 256×256 checkpoint and interpolating positional embeddings, followed by 5 epochs of training with batch sizes of 1024/512 for L/XL. Finally, we increase the resolution to 1024×1024 by loading the 512×512 checkpoint and again interpolating positional embeddings, and train for 2 epochs with batch sizes of 256/128 for L/XL. For a fair comparison, we also create a raster order counterpart following the same setup.

## B LOCALITY-AWARE GENERATION ORDER

### B.1 PYTORCH IMPLEMENTATION

```python
import numpy as np
import random

from scipy.spatial.distance import cdist
from scipy.spatial.distance import euclidean

def lpd_order_schedule(group_sizes=None, grid_size=16,
    proximity_threshold=1, repulsion_threshold=1):
    if group_sizes is None:
        group_sizes = [1] * (grid_size * grid_size)

    grid_coords = [[i, j] for i in range(grid_size) for j in
        range(grid_size)]
    selected_coords = []

    for step, group_size in enumerate(group_sizes):
        if step == 0:
            # For the first step, select a random coord. We always
                assume the group size for the first step is 1.
            selected_coords.append(random.choice(grid_coords))
            continue

        # Calculate the proximity score for all remaining grid coords
        candidates = []
        for coord in grid_coords:
            if coord in selected_coords:
                continue

            # Calculate the proximity score based on euclidean distance
                to already selected grid coords
            proximity_score = 0
            for selected_coord in selected_coords:
                if abs(coord[0] - selected_coord[0]) <= 1 and
                    abs(coord[1] - selected_coord[1]) <= 1:
                    distance = euclidean(coord, selected_coord)
                    if distance > 0:
                        proximity_score += 1.0 / distance
            candidates.append([proximity_score, coord])
```

```python
36          # Shuffle candidates so that grid coords with the same proximity
                score are randomly ordered
37          random.shuffle(candidates)
38          candidates.sort(key=lambda x: x[0], reverse=True)
39          candidates1 = [item[1] for item in candidates if item[0] >=
                proximity_threshold]
40          candidates2 = [item[1] for item in candidates if item[0] <
                proximity_threshold]
41
42          step_selected = []
43          step_filtered = []
44
45          # Proximity-based selection
46          while len(step_selected) < group_size and candidates1:
47              candidate = candidates1.pop(0)
48              too_close = False
49              for selected in step_selected:
50                  if abs(candidate[0] - selected[0]) <=
                        repulsion_threshold and abs(candidate[1] -
                        selected[1]) <= repulsion_threshold:
51                      too_close = True
52                      step_filtered.append(candidate)
53                      break
54
55              if not too_close:
56                  step_selected.append(candidate)
57
58          step_filtered.extend(candidates1)
59          candidates2.extend(step_filtered)
60
61          # Low-dependency selection
62          remaining = group_size - len(step_selected)
63          if remaining > 0:
64              step_selected.extend(farthest_point_sampling(step_selected,
                    candidates2, remaining))
65
66          selected_coords.extend(step_selected)
67
68      return np.ravel_multi_index(np.array(selected_coords).T, (grid_size,
            grid_size)).tolist()
69
70
71  def farthest_point_sampling(existing_points, candidate_points,
        num_to_select):
72      if len(candidate_points) <= num_to_select:
73          return candidate_points
74
75      # Convert to numpy arrays for efficient computation
76      existing_np = np.array(existing_points)
77      candidates_np = np.array(candidate_points)
78
79      # Initialize with existing points
80      selected_np = existing_np.copy()
81      selected_indices = []
82
83      for _ in range(num_to_select):
84          if len(selected_np) == 0:
85              # If no existing points, select randomly
86              idx = np.random.randint(len(candidates_np))
87              selected_np = candidates_np[idx][np.newaxis, :]
88          else:
89              # Calculate distances from all candidates to selected points
90              distances = cdist(candidates_np, selected_np)
91              min_distances = np.min(distances, axis=1)
92
```

```
93              # Set already selected candidates to 0 distance
94              min_distances[selected_indices] = 0
95
96              # Select the candidate with maximum minimum distance
97              idx = np.argmax(min_distances)
98              selected_np = np.vstack([selected_np, candidates_np[idx]])
99
100         selected_indices.append(idx)
101
102     return [candidate_points[i] for i in selected_indices]
```

## B.2 DISTINCTION FROM PREVIOUS WORK

The key distinction and primary advantage of our ordering mechanism is that we turn both principles into a single, explicit proximity objective. While previous works have observed each principle separately, none provide a way to quantify and jointly optimize them. In our method, we define a proximity metric that simultaneously (i) measures proximity to already generated context tokens and (ii) measures proximity among concurrently generated tokens, and we design an algorithm that optimizes generation orders with respect to both. For example, (Wang et al., 2024b) aim to reduce dependencies among concurrently generated tokens, but rely on a fixed region-wise parallel scheme, which inherently cannot both maximize proximity to previously generated tokens and minimize proximity within each concurrent group. Similarly, (Besnier et al., 2025) use a Halton-based ordering to decorrelate concurrent tokens; however, without a proximity metric their method cannot incorporate our first principle of staying close to existing context.

## B.3 SENSITIVITY ANALYSIS OF $\tau$ AND $\rho$

We conduct the sensitivity analysis using the LPD-XL model with 32 decoding steps on ImageNet 256×256. Our default configuration uses $\rho = 1$ and $\tau = 2$.

**Sensitivity analysis of $\tau$.** We fix $\rho = 1$ and vary $\tau$ to evaluate its impact (Table 6, left). When $\tau = 0$, no repulsion is applied, causing concurrently sampled tokens to be placed too close to one another and violating the requirement that concurrently generated tokens should maintain low proximity. Conversely, when $\tau$ is too large, concurrently sampled tokens are forced to be overly distant, making it difficult to find tokens that are both close to the already generated tokens and sufficiently far from each other. This violates the principle that newly generated tokens should preserve high proximity to existing tokens. Therefore, a moderate repulsion threshold is essential.

**Sensitivity analysis of $\rho$.** We fix $\tau = 2$ and vary $\rho$ to evaluate its impact (Table 6, right). A larger $\rho$ requires tokens to have very high proximity in order to be considered sufficiently close to the already generated tokens. This makes the condition increasingly difficult to satisfy—indeed, if $\rho$ were infinitely large, no token would meet the criterion, rendering the condition meaningless. Conversely, when $\rho = 0$, no threshold is imposed and all tokens are treated as sufficiently close to the generated ones. As shown in the results, the FID increases only slightly in this case. This is because our algorithm still sorts tokens by their computed proximity and selects them in descending order, so the first principle—prioritizing high-proximity tokens—remains intact. This also indicates that FID is relatively insensitive to smaller values of $\rho$.

Table 6: **Sensitivity analysis of $\tau$ and $\rho$.** All experiments use LPD-XL with 32 decoding steps on ImageNet 256×256. Default values ($\tau$=2, $\rho$=1) are underlined.

(a) Varying $\tau$ (fix $\rho = 1$)

| $\tau$ | 0 | 1 | 2 | 3 | 4 |
|---|---|---|---|---|---|
| FID↓ | 2.00 | 1.94 | **1.92** | 1.97 | 2.00 |

(b) Varying $\rho$ (fix $\tau = 2$)

| $\rho$ | 0 | 0.2 | 0.5 | 1 | 1.5 | 2 |
|---|---|---|---|---|---|---|
| FID↓ | 1.94 | 1.94 | 1.93 | **1.92** | 2.00 | 2.04 |

## C MORE VISUALIZATION OF ATTENTION MAPS

We provide partial visualization of the attention maps in Figure 2 and we provide more here. We select two layers each consists of 24 attention heads during the decoding and visualize them in Figure 10 and Figure 11.

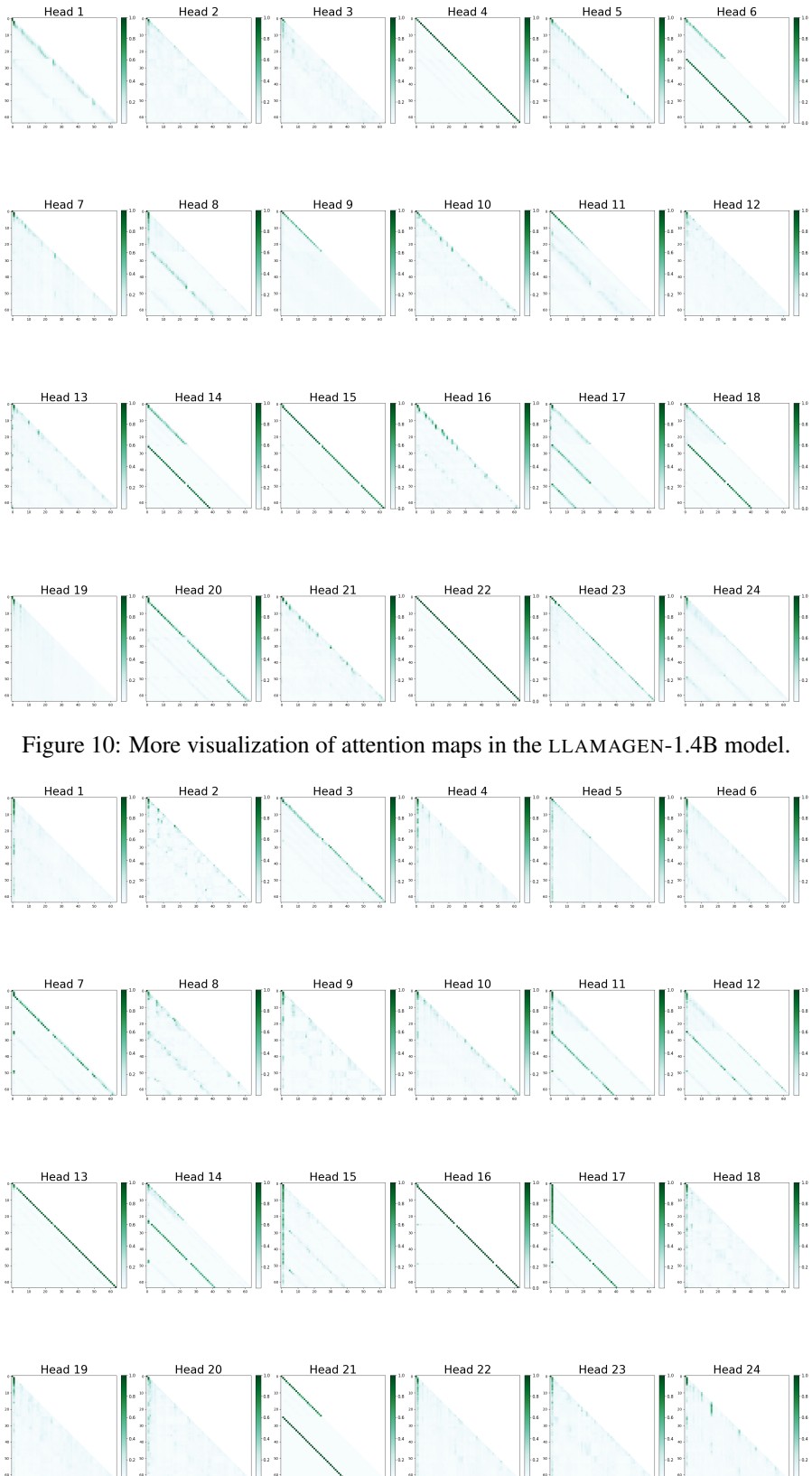

Figure 10: More visualization of attention maps in the LLAMAGEN-1.4B model.

Figure 11: More visualization of attention maps in the LLAMAGEN-1.4B model.

# D  MORE VISUALIZATION OF GENERATION EXAMPLES

Figure 12: **Generation Examples of Our Model.** We show 512×512 generation samples (top), 256×256 generation samples (middle) and zero-shot image editing results including class-conditional editing, inpainitng and outpainting (bottom).

Our model can naturally perform zero-shot editing tasks since we support image generation in arbitrary order. For image inpainting and outpainting, we prefill the KV cache with all tokens from the non-repaint regions along with a class token and generate the masked region in a random order. For class-conditional editing, we substitute the class embedding with a new class embedding and generate the edited region in a random order.

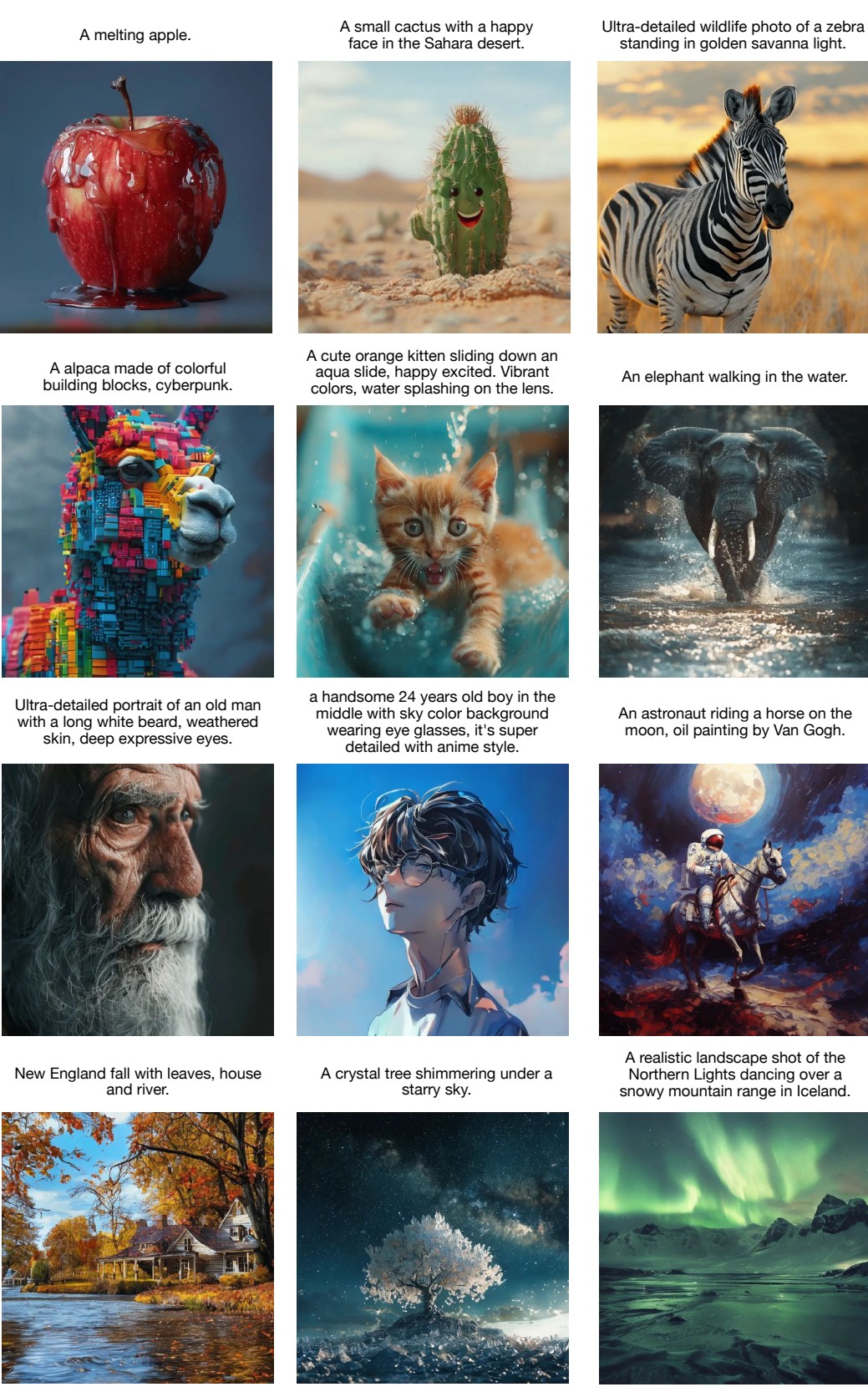

Figure 13: **Generation Examples of Our Model.** We show 1024×1024 text-to-image generation samples.

Table 7: Detailed per-category GenEval scores for text-to-image generation at 1024×1024 resolution.

| Model | Overall | Single Obj. | Two Obj. | Counting | Colors | Position | Color Attr. |
|---|---|---|---|---|---|---|---|
| Raster Counterpart-L | 0.55 | 0.94 | 0.67 | 0.39 | 0.78 | 0.16 | 0.36 |
| Raster Counterpart-XL | 0.60 | 0.99 | 0.71 | 0.46 | 0.82 | 0.25 | 0.36 |
| LPD-L | 0.58 | 1.00 | 0.66 | 0.39 | 0.85 | 0.17 | 0.42 |
| LPD-XL | 0.62 | 0.99 | 0.76 | 0.45 | 0.82 | 0.34 | 0.37 |

# E  EFFICIENCY ANALYSIS

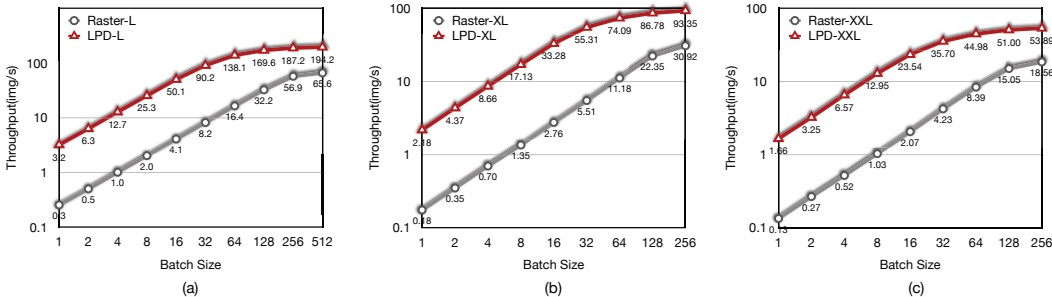

Figure 14: **Throughput vs. Batch Size on ImageNet 256×256 Class-Conditional Generation.** For LPD, we use 20 generation steps. Raster refers to the traditional fixed-raster-order generation model. We progressively increase the batch size until the process runs out of memory. The throughput values on the y-axis are plotted on a logarithmic scale.

As shown in Figure 14, LPD models are memory-bound when the batch size is 16 or smaller, as indicated by the linear increase in throughput with respect to batch size. When the batch size exceeds 16, the process gradually transitions from being memory-bound to compute-bound. For the traditional fixed-raster-order models, this transition occurs at a batch size around 128. Notably, when both models operate in the memory-bound regime, LPD consistently achieves nearly 12× higher throughput than the raster-order model—roughly matching the reduction in the number of generation steps. When at the maximum batch size, LPD still maintains a throughput advantage of approximately 3×.

