# OpenReview forum: "Locality-aware Parallel Decoding for Efficient Autoregressive Image Generation"
_ICLR.cc/2026/Conference — ICLR 2026 Oral_

### Official Review · Reviewer_axY1 · 2025-10-24

**Soundness:** 3
**Presentation:** 3
**Contribution:** 3
**Rating:** 8
**Confidence:** 4

**Summary:**

This paper proposes Locality-aware Parallel Decoding (LPD), an efficient method for parallel autoregressive image generation. By introducing position query tokens, it enables flexible, location-independent parallel decoding, while a locality-aware generation order improves consistency and quality. Experiments show that LPD achieves comparable or better image quality with up to 10× fewer generation steps and 3–4× faster decoding speed.

**Strengths:**

Here are the main strengths of the paper:

Significant Speedup – LPD greatly reduces the number of decoding steps (up to 10–20× fewer) while maintaining or improving image quality.

High-Quality Generation – The locality-aware scheduling preserves spatial coherence, producing consistent and detailed images even under high parallelism.

Flexible Decoding Framework – The position query token design allows generation in arbitrary orders, enabling diverse tasks like inpainting and outpainting without retraining.

Strong Empirical Validation – Extensive experiments on ImageNet (256×256 and 512×512) demonstrate clear improvements in both FID and latency over prior autoregressive baselines.

**Weaknesses:**

The paper is technically strong and the proposed method is clearly effective for autoregressive image generation. However, I think the evaluation could be further strengthened by extending it beyond ImageNet. In particular, it would be interesting to test the method on text-conditioned generation at higher resolutions (e.g., 1024²) to see whether the parallel decoding and locality assumptions still hold under more complex, long-range dependencies.
Moreover, applying LPD to video generation or temporal sequence modeling could highlight its scalability in spatiotemporal domains. Even a small-scale video experiment (e.g., UCF-101) would make the paper more convincing in terms of generality and impact.

**Questions:**

see above

---

> ### Author Response · Authors · 2025-11-26
> **Response to Reviewer axY1 (part 1)**
>
> Dear Reviewer axY1,
>
> Thank you so much for the careful review and insightful comments! We would like to provide more experiments support regarding your questions. We are happy to provide further information if you have additional questions.
>
> **W1: Text-conditioned image generation at higher resolutions (e.g. 1024^2).**
>
> Our method is general and can be readily extended to higher-resolution text-to-image generation (e.g., 1024×1024). Below, we provide a detailed description of this achievement in terms of model architecture, data, and training procedure. We then evaluate the resulting model on the widely used GenEval[1] benchmark to demonstrate the effectiveness and efficiency of our approach.
>
> **Model.**
> To enable the model to process user prompts rather than class labels, we incorporate the Gemma-2-2B[2] model as the text encoder, which converts prompts into text embeddings. These embeddings are passed through a linear projection layer and then concatenated with the image embeddings, with the text embeddings placed at the beginning of the sequence. All other components remain identical to our class-conditioned models. Our locality-aware generation-order schedule continues to apply, as only the spatial resolution needs to be increased.
>
> **Data.**
> We use an internal MidJourney-style synthetic dataset containing approximately 10M images. All images are re-captioned using Qwen2.5-VL-32B[3], which produces both a concise caption and a detailed caption for each image. During training, we randomly sample one of the two captions to serve as the text prompt.
>
> **Training Recipe.**
> We train two models, LPD-L (344M) and LPD-XL (760M), using a three-stage progressive-resolution schedule. We first train at 256×256 resolution for 40 epochs with batch sizes of 2048/1024 for the L/XL models. We then increase the resolution to 512×512 by loading the 256×256 checkpoint and interpolating positional embeddings, followed by 5 epochs of training with batch sizes of 1024/512 for L/XL. Finally, we increase the resolution to 1024×1024 by loading the 512×512 checkpoint and again interpolating positional embeddings, and train for 2 epochs with batch sizes of 256/128 for L/XL. For a fair comparison, we also create a raster order counterpart following the same setup.
>
> We test on the widely used GenEval[4] benchmark to evaluate the generation quality. We also measure the latency with a batch size of 1 and throughput with a batch size of 16 on a single NVIDIA A100 GPU under BFloat16 precision, with classifier-free guidance (CFG) for both.
>
> | Model     | #Param | #Steps | GenEval Score | Latency (s) | Throughput (img/s) |
> |-----------|--------|--------|----------------|-------------|---------------------|
> | Raster Counterpart-L  | 344M   | 4096   | 0.55           | 64.7        | 0.20                |
> | LPD-L     | 344M   | 64     | 0.58           | 1.01    | 5.28            |
> | Raster Counterpart-XL | 760M   | 4096   | 0.60           | 93.8        | 0.14                |
> | LPD-XL    | 760M   | 64     | 0.62           | 1.53    | 2.85            |
>
> We also provide the detailed GenEval score in the following table.
>
> | Model     | Overall | Single Object | Two Objects | Counting | Colors | Position | Color Attribution |
> |-----------|----------|----------------|--------------|----------|--------|----------|--------------------|
> | Raster Counterpart-L  | 0.55     | 0.94           | 0.67         | 0.39     | 0.78   | 0.16     | 0.36               |
> | LPD-L     | 0.58     | 1.00           | 0.66         | 0.39     | 0.85   | 0.17     | 0.42               |
> | Raster Counterpart-XL | 0.60     | 0.99           | 0.71         | 0.46     | 0.82   | 0.25     | 0.36               |
> | LPD-XL    | 0.62     | 0.99           | 0.76         | 0.45     | 0.82   | 0.34     | 0.37               |
>
> As demonstrated in the table, LPD reduces the sampling steps for 1024×1024 image generation from 4096 to just 64—a **64× decrease**—and simultaneously improves the GenEval score. This provides compelling evidence that LPD is a generalizable method capable of supporting high-resolution text-to-image generation. We also include qualitative generation results in Figure 13 in the appendix.
>
> [1] Ghosh, Dhruba, Hannaneh Hajishirzi, and Ludwig Schmidt. "Geneval: An object-focused framework for evaluating text-to-image alignment." Advances in Neural Information Processing Systems 36 (2023): 52132-52152.
>
> [2] Team, Gemma, et al. "Gemma 2: Improving open language models at a practical size." arXiv preprint arXiv:2408.00118 (2024).
>
> [3] Bai, Shuai, et al. "Qwen2. 5-vl technical report." arXiv preprint arXiv:2502.13923 (2025).

---

> ### Author Response · Authors · 2025-11-26
> **Response to Reviewer axY1 (part 2)**
>
> **W2: Applying LPD to video generation (e.g. UCF-101).**
>
> Our method can also definitely be extended to video generation. We verify this on the UCF-101[4] class to video generation task. We provide a detailed description of our implementation in terms of model architecture, data, and training procedure. We then conduct the evaluation using the Frechet Video Distance (FVD)[5] metric.
>
> **Model.**
> To encode videos, we use the LARP[6] video tokenizer, which maps a $16\times128\times128$ videos into $4\times16\times16$ tokens, which are then flattened into a 1D sequence. The change to the model architecture is minimal, as we only need to adjust the number of class embeddings to 101. For the locality-aware generation-order schedule, instead of calculating the proximity over the 2D grids, we just need to calculate it over the 3D grids taking the temporal dimension into consideration without any other modification.
>
> **Data.**
> We use the UCF-101[1] dataset, which comprises 101 action classes and approximately 13K videos. During training, we randomly sample a 16-frame clip from each video at every iteration.
>
> **Training Recipe.**
> We train two models, LPD-L (344M) and LPD-XL (760M), for 3000 epochs using batch sizes of 512 and 256, respectively. For a fair comparison, we also construct a raster-order counterpart under the same training setup.
>
> We use FVD[2] to evaluate generation quality. We also measure the latency with a batch size of 1 and throughput with a batch size of 64 on a single NVIDIA A100 GPU under BFloat16 precision, with classifier-free guidance (CFG) for both.
>
> | Model      | #Param | #Steps | FVD   | Latency (s) | Throughput (img/s) |
> |------------|--------|--------|-------|-------------|---------------------|
> | Raster Counterpart-L   | 344M   | 1024   | 102.3 | 14.3        | 3.81                |
> | LPD-L      | 344M   | 48     | 100.9 | 0.69    | 35.20         |
> | Raster Counterpart-XL  | 760M   | 1024   | 81.3  | 20.8        | 2.40                |
> | LPD-XL     | 760M   | 48     | 80.8  | 1.02    | 17.99          |
>
> As shown in the table, we reduce the number of generation steps from 1024 to 48 without sacrificing generation quality, leading to a **20× reduction** in latency. This demonstrates the effectiveness of our method in the video domain.
>
> [4] Soomro, Khurram, Amir Roshan Zamir, and Mubarak Shah. "Ucf101: A dataset of 101 human actions classes from videos in the wild." arXiv preprint arXiv:1212.0402 (2012).
>
> [5] Unterthiner, Thomas, et al. "Towards accurate generative models of video: A new metric & challenges." arXiv preprint arXiv:1812.01717 (2018).
>
> [6] Wang, Hanyu, et al. "Larp: Tokenizing videos with a learned autoregressive generative prior." arXiv preprint arXiv:2410.21264 (2024).
>
> Best,
>
> Authors

---

> > ### Comment · Reviewer_axY1 · 2025-11-26
> >
> > Thank you for your response. It addressed my concerns. I would like to keep the score.

---

### Official Review · Reviewer_mUfX · 2025-10-29

**Soundness:** 3
**Presentation:** 3
**Contribution:** 3
**Rating:** 6
**Confidence:** 5

**Summary:**

The paper introduces Locality-aware Parallel Decoding (LPD) with two techniques to accelerate autoregressive image generation. First, Flexible Parallelized Autoregressive Modeling leverages position-aware query tokens to indicate the tokens to be generated, enabling arbitrary generation ordering and degrees of parallelization. Second, a Locality-aware Generation Ordering is proposed to minimize mutual dependencies during parallel generation. Experiments on ImageNet 256$\times$256 and 512$\times$512 demonstrate the effectiveness of the proposed LPD.

**Strengths:**

1. The proposed Flexible Parallelized Autoregressive Modeling overcomes the constraint of a fixed generation order by allowing images to be synthesized in an arbitrary sequence. This capability holds the potential for discovering more effective generation orders in the future.
2. When equipped with the proposed Locality-aware Generation Ordering strategy, LPD demonstrates improved FID scores and greater generation efficiency on the ImageNet dataset.
3. The paper is easy to read and the figures are informative.

**Weaknesses:**

1. The paper's core algorithm (Algorithm 1) is presented in the appendix. While space constraints are understandable, the most critical algorithm should ideally be included in the main text, or at the very least, its underlying principles should be explained there.
2. The computational cost of the model increases compared to traditional fixed-order autoregressive models due to the use of additional positional query tokens. However, an analysis of this overhead is absent from the paper.
3. In line 208, the authors state that in previous methods, "tokens generated within the same parallel step are produced independently of one another." However, the paper does not rigorously analyze the issue of conditional independence in LPD's parallel sampling. It appears that while LPD ensures visibility among all target positions predicted concurrently, it may not fully resolve the underlying conditional independence of the parallel-generated tokens. A theoretical justification for this aspect would be beneficial.
4. The authors train the LPD model for 450 and 500 epochs on ImageNet at 256×256 and 512×512 resolutions, respectively. In contrast, most baseline methods (e.g., LlamaGen, PAR, and NAR) are trained for only 300 epochs. This discrepancy in training budgets may lead to an unfair comparison.

**Questions:**

See weaknesses.

---

> ### Author Response · Authors · 2025-11-26
> **Response to Reviewer mUfX**
>
> Dear Reviewer mUfX,
>
> Thank you so much for the careful review and valuable suggestions! We would like to provide more details regarding your questions. We hope our clarifications resolve your concerns, and we are happy to provide further information if you have additional questions.
>
> **W1: Include the core algorithm in the main text.**
>
> Thanks for your suggestion. We agree that the core technical algorithm should be presented in the main text. Accordingly, we have moved it from the appendix into the main paper (blue text).
>
> **W2: Analysis of the overhead for the use of additional positional query tokens.**
>
> While introducing positional query tokens increases the FLOPs, it does not materially affect inference latency. At inference time, the KV cache size is unchanged (we only cache sampled image tokens) and each decoding step remains memory-bound (as explained in the footnote on the first page). As illustrated in Figure 5, we fuse the encoding of sampled tokens with the decoding of new tokens into a single step. Consequently, the total number of decoding steps does not increase, and any additional compute has negligible impact on wall-clock latency.
>
> To further verify this, we use our model to simulate the traditional fixed raster-order generation—i.e., decoding exactly one token per step in raster order—while employing our positional query tokens. We then compare this setup with a standard fixed raster-order baseline and measure end-to-end latency on ImageNet 256×256. The results confirm that our positional query token does not introduce additional latency.
>
> | Model  | Latency(s) |
> |--------|------------|
> | Fixed Raster order-L | 3.73 |
> | LPD Simulate Raster-L | 3.75 |
> | Fixed Raster order-XL | 5.29 |
> | LPD Simulate Raster-XL | 5.29|
> | Fixed Raster order-XXL | 7.10 |
> | LPD Simulate Raster-XXL | 7.11 |
>
> **W3: A theoretical justification for the underlying conditional independence of the parallel-generated tokens.**
>
> We would like to clarify that **ensuring mutual visibility among concurrently generated tokens is orthogonal to the token dependency.**
>
> **Mutual visibility.** As shown in Figure 6, our specialized attention pattern guarantees visibility among concurrently generated tokens and is strictly more expressive than those used in prior work. In the worst case, it degrades to the attention pattern adopted by previous methods. Importantly, this mechanism is independent of the token dependency issue. Evidence for this is provided in Figure 9(a), where we use a random generation order. The results show that as the number of generation steps decreases, our model exhibits a significantly smaller FID degradation compared to the other two models. This demonstrates that our design effectively ensures mutual visibility among concurrent tokens, regardless of the ordering or dependencies among them.
>
> **Token dependency.** Our analysis indicates that tokens with higher spatial proximity tend to exhibit stronger dependencies. Motivated by this observation, we propose a locality-aware generation ordering to reduce dependencies among concurrently generated tokens. While we acknowledge that no ordering can completely eliminate such dependencies, our empirical results show that the proposed strategy substantially alleviates the issue.
>
> **W4: Discrepancy in training budgets may lead to an unfair comparison.**
>
> As shown in Tables 1 and 2, we also train the Raster-Counterpart models for 450 and 500 epochs, respectively. The comparison with the Raster-Counterpart baselines is fair. Following your suggestion, we additionally evaluate our method at 300 epochs, matching the training schedule of other baseline models (e.g., LlamaGen, PAR, and NAR). As stated in the paper, our training recipe uses a constant learning rate followed by a cosine decay over the final 50 epochs. Therefore, to obtain a 300-epoch checkpoint, we take the 250-epoch model and apply a 50-epoch cosine decay schedule. We summarize the results in the following table. The performance at 300 epochs is nearly identical to that at 450 epochs, indicating that 300 epochs are sufficient for the model to converge.
>
> | Model  | #Steps | FID (300 epochs) | FID (450 epochs) |
> |--------|--------|------------|------------|
> | LPD-L  | 20     |     2.44   |   2.40     |
> | LPD-XL | 20     |     2.10   |   2.10     |
> | LPD-XXL| 20     |     2.02   |   2.00     |
> | LPD-L  | 32     |     2.28   |   2.29     |
> | LPD-XL | 32     |     1.92   |   1.92     |
>
> Best,
>
> Authors

---

> > ### Comment · Reviewer_mUfX · 2025-11-27
> >
> > Thanks for the discussion. However, regarding W2, does the decoding process remain memory-bound regardless of the batch size?

---

> ### Author Response · Authors · 2025-11-28
> **Response to Reviewer mUfX**
>
> Thank you for your response! Regarding your follow-up question, we conducted a comprehensive evaluation of the throughput as the batch size increased. For clarity, we present the results in a figure included in the appendix. We kindly invite you to refer to Figure 14 in Appendix E.
>
> As shown in Figure 14, LPD models are memory-bound when the batch size is 16 or smaller, as indicated by the linear increase in throughput with respect to batch size. When the batch size exceeds 16, the process gradually transitions from being memory-bound to compute-bound. For the traditional fixed-raster-order models, this transition occurs at a batch size around 128. Notably, when both models operate in the memory-bound regime, LPD consistently achieves nearly 12× higher throughput than the raster-order model, roughly matching the reduction in the number of generation steps. When at the maximum batch size, LPD still maintains a throughput advantage of approximately 3×.
>
> We hope our response addresses your questions. We would be glad to provide further discussion or clarification if needed!

---

> > ### Comment · Reviewer_mUfX · 2025-11-28
> >
> > That is exactly what I meant. When the batch size is large, the process becomes compute-bound, and the additional overhead introduced by extra positional query tokens should be quantitatively analyzed, as they increase the FLOPs.

---

> ### Author Response · Authors · 2025-11-28
> **Response to Reviewer mUfX**
>
> Thank you again for the thoughtful follow-up! When the batch size becomes large, the decoding workload transitions from memory-bound to compute-bound, and in that regime the additional FLOPs from positional query tokens begin to matter.
>
> We agree that this trade-off is important and should be articulated explicitly with quantitative analysis. In response, **we have revised the paper and added an “Efficiency Analysis” subsection (highlighted in blue) in Section 2.4 in the main paper to analyze this in detail**. We sincerely appreciate your suggestion, which has helped us present a more complete discussion of efficiency.
>
> Please feel free to let us know if you have any further questions or suggestions and we would be glad to provide additional clarification.

---

### Official Review · Reviewer_Gaej · 2025-11-01

**Soundness:** 4
**Presentation:** 3
**Contribution:** 4
**Rating:** 8
**Confidence:** 3

**Summary:**

This paper proposes a method to accelerate AR visual generation, which is typically bottlenecked by sequential token prediction and memory bandwidth. It introduces two core innovations: (1) Flexible Parallelized Autoregressive Modeling, which allows arbitrary generation order and parallel prediction using learnable position query tokens and (2) Locality-aware Generation Ordering, a scheduling strategy that groups spatially related tokens to minimize dependencies and maximize contextual support. These techniques reduce generation steps dramatically and achieve at least 3.4× lower latency than prior parallel AR models without compromising image quality on ImageNet benchmarks.

**Strengths:**

The paper conducts a genuinely deep analysis of the key factors that affect both performance and generation quality in parallel autoregressive decoding (e.g., group size, dependency structure, attention visibility), and then turns those observations into a coherent, end-to-end parallelization method rather than a single heuristic component.

Through careful comparisons with recent parallel AR implementations (e.g., encoder–decoder style SAR/ARPG and decoder-only RANDAR), the authors show that their design, especially the “mutual visibility among concurrently generated tokens + cache only generated tokens” part, is not just faster but architecturally better motivated, and they support this with latency as well as quality numbers.

The paper is very readable: the motivation is clearly laid out, figures are aligned with the text (the attention-mask figures in particular), and the training vs. inference formulations are described in a way that makes reproduction and reimplementation realistic even for non-authors.

**Weaknesses:**

Experiments are limited to image generation; since current AR models are increasingly used for multimodal I/O (image–text, video tokens, layout, even audio tokens), it would strengthen the claim of “general AR parallelization” to show at least one non-image setting (e.g., CLIP-conditioned image tokens, image+text joint decoding, or video latents).

The paper does not compare against the newest AR acceleration lines such as speculative decoding, speculative Jacobi-style decoding, or draft/verify variants; even a small-scale experiment would help position LPD as complementary vs. strictly better.

While there is throughput analysis at batch 64, the work does not fully characterize memory consumption and scaling beyond that point; since the method introduces extra query tokens and fused encode–decode steps, reporting peak GPU memory and how it scales with batch size and resolution would make the efficiency story more convincing.

**Questions:**

Could the proposed LPD framework be extended or adapted to multimodal generation (e.g., image–text, video, or audio tokens)? Given its reliance on spatial locality, how might token dependencies behave for non-visual modalities?

How would LPD compare in efficiency and quality against speculative decoding or speculative Jacobi decoding? Including at least one experiment could clarify whether LPD complements or surpasses these methods.

Since the fusion of encoding and decoding steps introduces additional query tokens, could you quantify peak GPU memory usage and performance scaling with batch size and resolution (beyond batch 64)?

The method partially decouples context and generation tokens. Could the authors clarify how this affects the probabilistic consistency of the AR factorization (Eq. 2 in the paper)? In addition, the paper highlights speedups, but how does varying the group size quantitatively affect image fidelity and diversity?

---

> ### Author Response · Authors · 2025-11-26
> **Response to Reviewer Gaej (part 1)**
>
> Dear Reviewer Gaej,
>
> Thank you so much for the careful review and insightful comments! We would like to provide more details regarding your questions. We hope our clarifications resolve your concerns, and we are happy to provide further information if you have additional questions.
>
> **W1&Q1: Can LPD framework be extended to multimodal generation?**
>
> **Yes! LPD is a general framework and can be extended to other settings.** We first demonstrate the effectiveness of LPD framework on **video generation**. We verify this on the UCF-101[1] class to video generation task. We provide a detailed description of our implementation in terms of model architecture, data, and training procedure. We then conduct the evaluation using the Frechet Video Distance (FVD)[2] metric.
>
> **Model.**
> To encode videos, we use the LARP[3] video tokenizer, which maps a $16\times128\times128$ videos into $4\times16\times16$ tokens, which are then flattened into a 1D sequence. The change to the model architecture is minimal, as we only need to adjust the number of class embeddings to 101. For the locality-aware generation-order schedule, instead of calculating the proximity over the 2D grids, we just need to calculate it over the 3D grids taking the temporal dimension into consideration without any other modification.
>
> **Data.**
> We use the UCF-101[1] dataset, which comprises 101 action classes and approximately 13K videos. During training, we randomly sample a 16-frame clip from each video at every iteration.
>
> **Training Recipe.**
> We train two models, LPD-L (344M) and LPD-XL (760M), for 3000 epochs using batch sizes of 512 and 256, respectively. For a fair comparison, we also construct a raster-order counterpart under the same training setup.
>
> We use FVD[2] to evaluate generation quality. We also measure the latency with a batch size of 1 and throughput with a batch size of 64 on a single NVIDIA A100 GPU under BFloat16 precision, with classifier-free guidance (CFG) for both.
>
> | Model      | #Param | #Steps | FVD   | Latency (s) | Throughput (img/s) |
> |------------|--------|--------|-------|-------------|---------------------|
> | Raster Counterpart-L   | 344M   | 1024   | 102.3 | 14.3        | 3.81                |
> | LPD-L      | 344M   | 48     | 100.9 | 0.69    | 35.20         |
> | Raster Counterpart-XL  | 760M   | 1024   | 81.3  | 20.8        | 2.40                |
> | LPD-XL     | 760M   | 48     | 80.8  | 1.02    | 17.99          |
>
> As shown in the table, we reduce the number of generation steps from 1024 to 48 without sacrificing generation quality, leading to a **20× reduction** in latency. This demonstrates the effectiveness of our method in the video domain.
>
> [1] Soomro, Khurram, Amir Roshan Zamir, and Mubarak Shah. "Ucf101: A dataset of 101 human actions classes from videos in the wild." arXiv preprint arXiv:1212.0402 (2012).
>
> [2] Unterthiner, Thomas, et al. "Towards accurate generative models of video: A new metric & challenges." arXiv preprint arXiv:1812.01717 (2018).
>
> [3] Wang, Hanyu, et al. "Larp: Tokenizing videos with a learned autoregressive generative prior." arXiv preprint arXiv:2410.21264 (2024).

---

> ### Author Response · Authors · 2025-11-26
> **Response to Reviewer Gaej (part 2)**
>
> We also demonstrate that the LPD framework can handle complex **text-to-image generation at a resolution of 1024×1024**. We provide a detailed description of this achievement in terms of model architecture, data, and training procedure. We then evaluate the resulting model on the widely used GenEval[4] benchmark to demonstrate the effectiveness and efficiency of our approach.
>
> **Model.**
> To enable the model to process user prompts rather than class labels, we incorporate the Gemma-2-2B[5] model as the text encoder, which converts prompts into text embeddings. These embeddings are passed through a linear projection layer and then concatenated with the image embeddings, with the text embeddings placed at the beginning of the sequence. All other components remain identical to our class-conditioned models. Our locality-aware generation-order schedule continues to apply, as only the spatial resolution needs to be increased.
>
> **Data.**
> We use an internal MidJourney-style synthetic dataset containing approximately 10M images. All images are re-captioned using Qwen2.5-VL-32B[6], which produces both a concise caption and a detailed caption for each image. During training, we randomly sample one of the two captions to serve as the text prompt.
>
> **Training Recipe.**
> We train two models, LPD-L (344M) and LPD-XL (760M), using a three-stage progressive-resolution schedule. We first train at 256×256 resolution for 40 epochs with batch sizes of 2048/1024 for the L/XL models. We then increase the resolution to 512×512 by loading the 256×256 checkpoint and interpolating positional embeddings, followed by 5 epochs of training with batch sizes of 1024/512 for L/XL. Finally, we increase the resolution to 1024×1024 by loading the 512×512 checkpoint and again interpolating positional embeddings, and train for 2 epochs with batch sizes of 256/128 for L/XL. For a fair comparison, we also create a raster order counterpart following the same setup.
>
> We test on the widely used GenEval[4] benchmark to evaluate the generation quality. We also measure the latency with a batch size of 1 and throughput with a batch size of 16 on a single NVIDIA A100 GPU under BFloat16 precision, with classifier-free guidance (CFG) for both.
>
> | Model     | #Param | #Steps | GenEval Score | Latency (s) | Throughput (img/s) |
> |-----------|--------|--------|----------------|-------------|---------------------|
> | Raster Counterpart-L  | 344M   | 4096   | 0.55           | 64.7        | 0.20                |
> | LPD-L     | 344M   | 64     | 0.58           | 1.01    | 5.28            |
> | Raster Counterpart-XL | 760M   | 4096   | 0.60           | 93.8        | 0.14                |
> | LPD-XL    | 760M   | 64     | 0.62           | 1.53    | 2.85            |
>
> We also provide the detailed GenEval score in the following table.
>
> | Model     | Overall | Single Object | Two Objects | Counting | Colors | Position | Color Attribution |
> |-----------|----------|----------------|--------------|----------|--------|----------|--------------------|
> | Raster Counterpart-L  | 0.55     | 0.94           | 0.67         | 0.39     | 0.78   | 0.16     | 0.36               |
> | LPD-L     | 0.58     | 1.00           | 0.66         | 0.39     | 0.85   | 0.17     | 0.42               |
> | Raster Counterpart-XL | 0.60     | 0.99           | 0.71         | 0.46     | 0.82   | 0.25     | 0.36               |
> | LPD-XL    | 0.62     | 0.99           | 0.76         | 0.45     | 0.82   | 0.34     | 0.37               |
>
> As demonstrated in the table, LPD reduces the sampling steps for 1024×1024 image generation from 4096 to just 64—a **64× decrease**—and simultaneously improves the GenEval score. This provides compelling evidence that LPD is a generalizable method capable of supporting high-resolution text-to-image generation. We also include qualitative generation results in Figure 13 in the appendix.
>
> [4] Ghosh, Dhruba, Hannaneh Hajishirzi, and Ludwig Schmidt. "Geneval: An object-focused framework for evaluating text-to-image alignment." Advances in Neural Information Processing Systems 36 (2023): 52132-52152.
>
> [5] Team, Gemma, et al. "Gemma 2: Improving open language models at a practical size." arXiv preprint arXiv:2408.00118 (2024).
>
> [6] Bai, Shuai, et al. "Qwen2. 5-vl technical report." arXiv preprint arXiv:2502.13923 (2025).

---

> ### Author Response · Authors · 2025-11-26
> **Response to Reviewer Gaej (part 3)**
>
> **W2&Q2: How would LPD compare in efficiency and quality against speculative decoding or speculative Jacobi decoding?**
>
> We integrate Speculative Jacobi Decoding (SJD)[7] into our Raster Counterpart-XL model, referred to as Raster Counterpart-XL + SJD. Recall that Raster Counterpart-XL is the raster-order baseline described in the paper and reported in Table 1. Following the same evaluation protocol, we conduct experiments on ImageNet 256×256 using a single NVIDIA A100 GPU with BFloat16 precision. Because the accepted token length varies across samples, we report the average number of decoding steps and latency for Raster Counterpart-XL + SJD. The corresponding results are presented in the table below.
>
> | Method                       | Steps  | FID  | Latency (s) |
> |------------------------------|--------|------|--------------|
> | Raster Counterpart-XL        | 256    | 2.12 | 5.41         |
> |  Raster Counterpart-XL  + SJD| 120.95 | 2.18 | 2.68         |
> | LPD-XL                       | 32     | 1.92 | 0.66         |
> | LPD-XL                       | 20     | 2.10 | 0.41         |
>
> SJD approximately halves both the number of decoding steps and the latency of the raster-order baseline, with a slight increase in FID. However, our LPD remains substantially more efficient while simultaneously achieving better FID. This indicates that LPD significantly outperforms speculative-decoding–style approaches.
>
> **W3&Q3: Quantify peak GPU memory usage and performance scaling with batch size and resolution.**
>
> We report the detailed peak GPU memory usage and the throughput measurement under different batch sizes (beyond 64) and resolutions.
>
> **256×256 Resolution-20 Steps**
>
> | Model  | Batch Size | Throughput (img/s)   | Peak Memory (GB) |
> |--------|-------------|---------------------|-------------------|
> | LPD-L  | 64          | 139.11              | 7.05              |
> | LPD-L   | 96          | 156.91              | 8.84              |
> | LPD-L  | 128         | 169.55              | 10.62             |
> | LPD-L   | 192         | 181.02              | 14.19             |
> | LPD-L  | 256         | 187.24              | 17.80             |
> | LPD-L   | 384         | 193.16              | 24.92             |
> |LPD-L   | 448         | 193.59              | 28.48             |
> |LPD-L  | 512         | 194.23              | 32.03             |
>
> | Model  | Batch Size | Throughput (img/s) | Peak Memory (GB) |
> |--------|-------------|---------------------|-------------------|
> | LPD-XL  | 64          | 75.20               | 13.65             |
> | LPD-XL  | 96          | 81.81               | 16.77             |
> | LPD-XL  | 128         | 86.44               | 19.89             |
> | LPD-XL | 192         | 90.97               | 26.15             |
> | LPD-XL | 256         | 93.16               | 32.46             |
> | LPD-XL  | 384         | 95.29               | 44.88             |
>
> | Model  | Batch Size | Throughput (img/s) | Peak Memory (GB) |
> |--------|-------------|---------------------|-------------------|
> | LPD-XXL| 64          | 45.07               | 23.20             |
> | LPD-XXL| 96          | 48.50               | 28.03             |
> | LPD-XXL| 128         | 50.87               | 32.87             |
> | LPD-XXL| 192         | 52.83               | 42.59             |
> | LPD-XXL| 256         | 53.55               | 52.24             |
>
> **256×256 Resolution-32 Steps**
>
> | Model  | Batch Size | Throughput (img/s) | Peak Memory (GB) |
> |--------|------------|---------------------|-------------------|
> |LPD-L   | 64         | 110.34              | 6.88              |
> |LPD-L  | 96         | 126.19              | 8.58              |
> |LPD-L  | 128        | 141.75              | 10.28             |
> |LPD-L  | 192        | 157.63           |   13.67          |
> | LPD-L | 256        | 164.91              | 17.07             |
> |LPD-L  | 384        | 173.59              | 23.86             |
> |LPD-L  | 448        | 175.46              | 27.26             |
> |LPD-L  | 512        | 176.93              | 30.65             |
>
> | Model  | Batch Size | Throughput (img/s) | Peak Memory (GB) |
> |--------|------------|---------------------|-------------------|
> | LPD-XL | 64         | 61.24               | 13.47             |
> | LPD-XL| 96         | 69.48               | 16.51             |
> | LPD-XL| 128        | 74.18               | 19.54             |
> | LPD-XL | 192        | 80.54               | 25.62             |
> | LPD-XL| 256        | 83.49               | 31.71             |
> |LPD-XL| 384        | 86.63               | 43.82             |
>
> **512×512 Resolution-48 Steps**
>
> | Model   | Batch Size | Throughput (img/s) | Peak Memory (GB) |
> |-|-|-|-|
> | LPD-L| 64 | 35.16  | 17.02|
> | LPD-L| 80| 36.32| 20.33|
> | LPD-L| 96 | 37.41| 23.64|
> |LPD-L| 128| 39.00| 30.27|
>
> |Model|Batch Size|Throughput (img/s)|Peak Memory (GB)|
> |-|-|-|-|
> |LPD-XL| 64|18.18| 31.60|
> |LPD-XL| 80|18.58| 37.57|
> |LPD-XL| 96| 19.03| 43.55|

---

> ### Author Response · Authors · 2025-11-26
> **Response to Reviewer Gaej (part 4)**
>
> **Q4.1: How would ​​decoupling context and generation tokens affect the probabilistic consistency of the AR factorization (Eq. 2 in the paper)?**
>
> Decoupling context and generation tokens does not alter the probabilistic consistency of the AR factorization. As described in Section 2.2 and illustrated in Figures 4 and 5, our attention mechanism guarantees that at each generation step the group $X_g$ is conditioned on all previously generated tokens $X_{<g}$. The only difference introduced by decoupling is that we use position query tokens as the input at each generation step, rather than the previously decoded tokens.
>
> **Q4.2: How does varying the group size quantitatively affect image fidelity and diversity?**
>
> Varying the group size equals to change the number of generation steps, since we increase the number of tokens generated per step via a cosine schedule. These results are already presented in Figures 9(a) and 9(b). Figure 9(a) shows how performance varies with different numbers of generation steps without our locality-aware parallel generation order schedule—that is, using only our flexible parallelized autoregressive modeling. Figure 9(b) shows the corresponding performance with our locality-aware parallel generation order schedule. The results show that reducing the number of generation steps has minimal impact on performance initially, but performance begins to degrade once the steps are reduced beyond 64/32.
>
> Best,
>
> Authors

---

### Official Review · Reviewer_z7UP · 2025-11-02

**Soundness:** 3
**Presentation:** 2
**Contribution:** 3
**Rating:** 6
**Confidence:** 4

**Summary:**

The paper introduces Locality-aware Parallel Decoding (LPD), a method designed to speed up autoregressive image generation. LPD uses two key techniques—Flexible Parallelized Autoregressive Modeling for arbitrary parallelization and Locality-aware Generation Ordering for quality enhancement—and reduces generation steps significantly (e.g., 256 to 20 for 256×256 images) on ImageNet while cutting latency by at least 3.4× compared to prior parallel models.

**Strengths:**

* Sound Method Design：The learnable position query tokens decouple context modeling from decoding, enabling generation at arbitrary target positions and boosting flexibility. The exploration of two locality principles, particularly the second one, offers meaningful insights for the community.

* Strong Performance：The method achieves clear reductions in generation steps and latency with good quality.

**Weaknesses:**

1. Overclaimed Contributions in Writing

   - For "Flexible Parallelized Autoregressive Modeling", decoder-only works like PAR/ZipAR/NAR already treat previously decoded tokens as KV Cache and the queries are decoded in parallel ensuring the mutual visibility among tokens generated concurrently; the key difference lies only in LPD’s position query tokens (enabling arbitrary target positions), which should be clarified to avoid overstating contributions.
   - For "Locality-aware Generation Ordering", the first principle is well-studied (Sec. 3.2 descriptions can be simplified), while the second principle ("low proximity among concurrent tokens") has been explored by Wang et al. (2024b) and Besnier et al. (2025)—its underexploration in prior work needs more detailed explanation in Sec. 3.2.

2. Insufficient Ablation Studies
   - Sensitivity analysis of critical thresholds $\tau$ and $\rho$ is missing, despite their importance to query position performance.
   - A key ablation is absent: for Flexible Parallelized AR Modeling (excluding adaptive generation order), testing whether replacing query tokens with LPD’s "position query tokens" would clarify the source of performance gains.

3. Confusing Details
    - Inference process ambiguity: It is unclear if sampled tokens need a forward pass to store KV Cache, and if this can be done in the final decoding step.
   - Figure 3 confusion: The figure shows queries attending only to the latest decoded tokens, conflicting with the claim that queries causally attend to all previously generated tokens.

4. Minor Structural Issue：The dynamic generation order, a core technical contribution, should be included in the main text.

**Questions:**

See weaknesses

---

> ### Author Response · Authors · 2025-11-26
> **Response to Reviewer z7UP (part 1)**
>
> Dear Reviewer z7UP,
>
> Thank you so much for the careful review and valuable suggestions! We would like to provide more details regarding your questions. We hope our clarifications resolve your concerns, and we are happy to provide further information if you have additional questions.
>
> **W1.1: Overclaimed for "Flexible Parallelized Autoregressive Modeling".**
>
> We respectfully disagree that our claims about the Flexible Parallelized Autoregressive Modeling architecture are overstated. We admit that methods like PAR/NAR/ZipAR ensure mutual visibility among tokens generated concurrently, but our contribution lies in **simultaneously** enabling arbitrary target position generation with flexible parallelization **and** preserving mutual visibility among tokens generated concurrently. It is not a trivial add-ons to previous architectures to support these two features simultaneously, which motivates us to develop the new architecture. **To the best of our knowledge, no prior work has simultaneously realized both capabilities within a single architecture.** Specifically: (1) Methods such as PAR/NAR/ZipAR adopt a predefined fixed generation order and maintain mutual visibility among concurrently generated tokens, yet they cannot support arbitrary target position generation with flexible parallelization. (2) Methods such as SAR/ARPG/RandAR do allow arbitrary target position generation, but the tokens generated in parallel are conditionally independent, lacking mutual visibility. These observations highlight inherent limitations in prior architectures. Only our learnable position query token combined with the proposed context–query attention mechanism can jointly support these two capabilities within a single architecture.These comparisons are presented in detail in Section 2.2 under “Comparison with Other Methods.” For additional clarity, we have revised the third paragraph (blue text) to more clearly differentiate our approach from PAR, NAR, and ZipAR.
>
> **W1.2: For "Locality-aware Generation Ordering", its underexploration for the second principle in prior work needs more detailed explanation.**
>
> The key distinction and primary advantage of our ordering mechanism is that we turn both principles into a single, explicit **proximity objective**. While previous works have observed each principle separately, none provide a way to quantify and jointly optimize them. In our method, we define a proximity metric that simultaneously (i) measures proximity to already generated context tokens and (ii) measures proximity among concurrently generated tokens, and we design an algorithm that optimizes generation orders with respect to both.
>
> For example, Wang et al.[1] aim to reduce dependencies among concurrently generated tokens, but rely on a fixed region-wise parallel scheme, which inherently cannot both maximize proximity to previously generated tokens and minimize proximity within each concurrent group. Similarly, Besnier et al.[2] use a Halton-based ordering to decorrelate concurrent tokens; however, without a proximity metric their method cannot incorporate our first principle of staying close to existing context. We have clarified this distinction in Section 2.3 (blue text).
>
> [1] Wang, Yuqing, et al. "Parallelized autoregressive visual generation." Proceedings of the Computer Vision and Pattern Recognition Conference. 2025.
>
> [2] Besnier, Victor, et al. "Halton scheduler for masked generative image transformer." arXiv preprint arXiv:2503.17076 (2025).

---

> ### Author Response · Authors · 2025-11-26
> **Response to Reviewer z7UP (part 2)**
>
> **W2.1: Sensitivity analysis of $\tau$ and $\rho$.**
>
> We conduct the sensitivity analysis using the LPD-XL model with 32 decoding steps on ImageNet 256×256, consistent with the ablation setting in the main paper. In the main paper, our default configuration uses $\rho$=1 and $\tau$=2.
>
> 1. **Sensitivity analysis of $\tau$.** We fix $\rho$=1 and vary $\tau$ to evaluate its impact. When $\tau$=0, no repulsion is applied, causing concurrently sampled tokens to be placed too close to one another and violating the requirement that concurrently generated tokens should maintain low proximity. Conversely, when $\tau$ is too large, concurrently sampled tokens are forced to be overly distant, making it difficult to find tokens that are both close to the already generated tokens and sufficiently far from each other. This violates the principle that newly generated tokens should preserve high proximity to existing tokens. Therefore, a moderate repulsion threshold is essential.
>
>     | $\tau$ = 0 | $\tau$ = 1 | $\tau$ = 2 | $\tau$ = 3 | $\tau$ = 4 |
>     |-------|-------|-------|-------|-------|
>     | 2.00  | 1.94  | 1.92  | 1.97  | 2.00  |
>
> 2. **Sensitivity analysis of $\rho$.** We fix $\tau$=2 and vary $\rho$ to evaluate its impact. A larger $\rho$ requires tokens to have very high proximity in order to be considered sufficiently close to the already generated tokens. This makes the condition increasingly difficult to satisfy—indeed, if $\rho$ were infinitely large, no token would meet the criterion, rendering the condition meaningless. Conversely, when $\rho$=0, no threshold is imposed and all tokens are treated as sufficiently close to the generated ones. As shown in the results, the FID increases only slightly in this case. This is because our algorithm still sorts tokens by their computed proximity and selects them in descending order, so the first principle—prioritizing high-proximity tokens—remains intact. This also indicates that FID is relatively insensitive to smaller values of $\rho$.
>
>     | $\rho$ = 0 | $\rho$ = 0.2 | $\rho$ = 0.5 | $\rho$ = 1 | $\rho$ = 1.5 | $\rho$ = 2 |
>     |-------|---------|---------|-------|---------|-------|
>     | 1.94  | 1.94    | 1.93    | 1.92  | 2.00    | 2.04  |
>
> **W2.2: Ablation for replacing query tokens with LPD’s "position query tokens".**
>
> We think you are referring to “query tokens’’ in the context of the conventional raster-order model. To verify this, we conduct an experiment using the XL-sized model on ImageNet 256×256. We train (1) a standard raster-order model using the original query tokens (actually the last decoded token) and (2) our LPD model using the proposed position query tokens, while disabling parallel generation—in both cases generating one token per step in raster order. As shown in the table below, both approaches achieve identical FID scores, indicating comparable modeling capacity. The effectiveness of our position query tokens becomes evident when the number of generation steps is reduced. As shown in Figure 9(a), when the generation steps decrease, our model exhibits a substantially smaller rise in FID compared with the other two models.
>
> |Type                  |FID |
> |----------------------|----|
> |Query Token           |2.12|
> |Positional Query Token|2.12|
>
> **W3.1: Inference process ambiguity.**
>
> The sampled tokens require a forward pass to store the KV cache, which is needed to provide context for subsequently decoded tokens. In the final decoding step, however, no forward pass is required for the sampled tokens because they are the last tokens generated and no longer need to provide context for future decoding.
>
> **W3.2: Confusion about Figure 3.**
>
> The tokens grouped in the same dashed box indicate only that they are provided as input to the model at the same step. They do not represent the set of tokens that the queries only attend to. Figure 3 is intended primarily to illustrate the model’s input–output structure and to enable a clear comparison with the traditional raster generation process. The attention mechanisms are shown in Figures 4 and 5, where the attention masks explicitly demonstrate that “queries causally attend to all previously generated tokens.”
>
> **W4: Include the generation order algorithm in the main text.**
>
> Thanks for your suggestion. We agree that the core technical algorithm should be presented in the main text. Accordingly, we have moved it from the appendix into the main paper (blue text).
>
> Best,
>
> Authors

---

### Author Response · Authors · 2025-11-26
**Global Response**

We appreciate all the reviewers for their insightful and constructive feedback. We are glad that many of the strengths of our paper have been recognized and appreciated by the reviewers:

- **Flexibility in Generation Order** – Learnable position query tokens decouple context modeling from decoding, enabling arbitrary generation orders while ensuring mutual visibility among concurrently generated tokens.
- **Effective Locality Modeling** – The locality-aware generation strategy improves spatial coherence and boosts image quality.
- **Significant Speedup with Maintained Quality** – The method achieves substantial reductions in decoding steps and latency while keeping or improving output quality.
- **Well-Motivated Architectural Design** – Parallelization design choices (e.g., mutual visibility, caching only generated tokens) are principled and empirically validated against prior parallel AR approaches.
- **Thorough Analysis & Strong Empirical Evidence** – The paper provides deep analysis of key factors affecting parallel AR performance and offers comprehensive experiments demonstrating clear advantages.
- **Clarity & Reproducibility** – The paper is clearly written with intuitive figures and detailed methodology, making it easy for others to understand and reproduce.

We have responded separately to each reviewer to address their specific questions. In addition to these individual results, we have also validated the effectiveness of our LPD framework on **1024×1024 text-to-image generation** and **class-to-video generation** as suggested by Reviewer Gaej and axY1. If you are intersted about the details, please kindly refer to our response to Reviewer axY1. We believe these two additional experiments provide strong further evidence that LPD is a general and effective framework.

---

### Author Response · Authors · 2025-12-04
**Rebuttal Summary for AC**

Dear AC,

Thank you for taking the time to review our paper! We would like to provide a brief rebuttal summary for your convenience.

**1. Summary of Reviewer Strengths.**

- **Flexibility in Generation Order** (Reviewer z7UP, mUfX, axY1) – Learnable position query tokens decouple context modeling from decoding, enabling arbitrary generation orders while ensuring mutual visibility among concurrently generated tokens.
- **Effective Locality Modeling** (Reviewer z7UP, mUfX) – The locality-aware generation strategy improves spatial coherence and boosts image quality. It also offers meaningful insights for the community.
- **Significant Speedup with Maintained Quality** (Reviewer z7UP, mUfX, axY1) – The method achieves substantial reductions in decoding steps and latency while keeping or improving output quality.
- **Well-Motivated Architectural Design** (Reviewer Gaej)– Parallelization design choices (e.g., mutual visibility, caching only generated tokens) are principled and empirically validated against prior parallel AR approaches.
- **Thorough Analysis & Strong Empirical Evidence** (Reviewer Gaej, axY1) – The paper provides deep analysis of key factors affecting parallel AR performance and offers comprehensive experiments demonstrating clear advantages.
- **Clarity & Reproducibility** (Reviewer Gaej, mUfX)– The paper is clearly written with intuitive figures and detailed methodology, making it easy for others to understand and reproduce.

**2. Response Summary to Each Reviewer’s Concern.**

1) Reviewer z7UP (Rating 6):

    **Main Concern:** Overclaimed Contribution; Insufficient Ablation Studies

    **Our Response:** (1) **We have not overstated our contributions.** We are the first work to **simultaneously** enabling arbitrary target position generation with flexible parallelization and preserving mutual visibility among tokens generated concurrently. We are the first work to propose a **single, explicit proximity metric to quantify and jointly optimize the two locality principles**. (2) We provide all the requested ablation studies along with a comprehensive analysis.

2) Reviewer Gaej (Rating 8):

    **Main Concern:** Extend LPD to multimodal generation; Comparison with speculative decoding; Peak GPU memory usage and throughput scaling with batch size and resolution

    **Our Response:** (1) We extend the LPD framework to **1024×1024 text-to-image generation** and **class-to-video generation**, achieving **64×** and **20×** latency reductions compared to the raster-order baseline, respectively, while maintaining or improving generation quality. (2) We compare our approach with the state-of-the-art speculative decoding method and show that the speculative decoding reduces generation steps by **only 2×**, whereas our method achieves a **12×** reduction without degrading quality. (3) We report the detailed peak GPU memory usage and the throughput measurement under different batch sizes and resolutions.

3) Reviewer mUfX (Rating 6):

    **Main Concern:** Analysis of the increased computational cost is absent; A theoretical justification for the underlying conditional independence; Discrepancy in training budgets with other methods

    **Our Response:** (1) We **add a new subsection** "Efficiency Analysis" in the paper to thoroughly discuss the increased computational cost. In memory-bound settings such as small-batch inference, it has a **negligible impact on wall-clock latency**. As the batch size increases, the system progressively shifts toward a compute-bound regime, where it begins to matter and diminish the speedup. But even at the maximum feasible batch size, our method retains a throughput advantage of approximately **3×** over the raster-order baseline. In Table 1, we also report the throughput of all models using a reasonably large batch size of 64. Our model achieves **substantially higher throughput** than the others, demonstrating its efficiency. (2) We clarify that ensuring mutual visibility among concurrently generated tokens is **orthogonal** to the token dependency. (3) We evaluate our method at 300 epochs, matching the training schedule of other baseline models. The performance at 300 epochs is nearly identical to that at 450 epochs, indicating that **300 epochs are sufficient for the model to converge**, which doesn't affect the results of our comparison with others.

4) Reviewer axY1 (Rating 8):

    **Main Concern:**  Extend the method on text-conditioned generation at higher resolutions (1024²); Apply LPD to video generation

    **Our Response:** (1) We extend the LPD framework to **1024×1024 text-to-image generation**, achieving a **64×** latency reduction compared to the raster-order baseline while improves the GenEval score. Our 760M model generates a high-quality 1024×1024 image in **just 1.53 seconds** on a single A100 GPU. (2) We apply LPD to **class-to-video generation**, achieving **20×** latency reduction compared to the raster-order baseline with maintained quality.

---

### Meta-Review · Area_Chair_pE1x · 2025-12-25

**Summary:**

This paper proposes Locality-aware Parallel Decoding (LPD), a framework designed to accelerate autoregressive (AR) image generation. The method introduces two key innovations: (1) Flexible Parallelized Autoregressive Modeling, which utilizes learnable position query tokens to enable arbitrary generation ordering and parallelization, and (2) Locality-aware Generation Ordering, a technique that minimizes intra-group dependencies while enhancing generation quality. The empirical results demonstrate that LPD reduces generation steps significantly (e.g., from $256$ to $20$ for $256\times256$ images) and achieves at least $3.4\times$ lower latency compared to previous parallel AR models, while maintaining or improving output quality.

The main concerns raised by the reviewers focused on the scope and depth of evaluation rather than the core technical soundness of the proposed method. Specifically, reviewers questioned (i) whether the claimed contributions were sufficiently distinguished from prior parallel autoregressive decoding methods, (ii) whether key design choices—such as position query tokens, locality-aware generation ordering, and threshold parameters—were adequately ablated and analyzed, and (iii) whether the empirical validation was limited by focusing primarily on ImageNet image generation without comparisons to newer AR acceleration techniques or extensions to multimodal settings. None of the reviewers raised concerns about correctness, empirical validity, or reproducibility. The overall sentiment across reviews was that the paper proposes a strong and well-motivated method with clear empirical benefits, but that additional analysis and broader evaluation would be needed to fully establish its generality and impact. These concerns, combined with multiple positive assessments highlighting novelty and strong performance, informed my decision to recommend acceptance.

**Reviewer Concerns:**

All concerns raised by reviewers were addressed during the discussion:

+ Overclaimed contributions and novelty clarification (Reviewer z7UP): The authors clearly differentiated their approach from prior parallel AR methods, emphasizing that LPD is the first to simultaneously support arbitrary target position generation and mutual visibility among concurrently generated tokens within a single architecture.
+ Missing ablations and sensitivity analyses (Reviewer z7UP, mUfX): The rebuttal includes comprehensive ablations on key thresholds and isolates the effect of position query tokens, substantially clarifying the sources of performance gains.

+ Algorithm placement and clarity (Reviewer z7UP, mUfX): The core generation-order algorithm has been moved into the main text, resolving concerns about accessibility of the central method.
+ Efficiency, memory, and training budget fairness (Reviewer Gaej, mUfX): Detailed throughput, memory scaling, and matched-training-budget experiments were added, demonstrating that the additional query tokens do not undermine efficiency and that comparisons are fair.
+ Scope limitations (Reviewer Gaej, axY1): The authors extended LPD to high-resolution ($1024\times1024$) text-to-image generation and to video generation, providing strong evidence that the framework generalizes beyond ImageNet images.
+ Comparison with speculative decoding (Reviewer Gaej): Direct experimental comparisons show that LPD significantly outperforms speculative decoding–style approaches in both speed and quality.

**Reviewer Scores:**

The authors' rebuttal has satisfactorily addressed major concerns that could affect acceptance. All reviewers express either explicit acceptance recommendations or borderline-positive assessments.

+ Reviewer z7UP: $6\rightarrow6$.
+ Reviewer Gaej: $8\rightarrow8$.
+ Reviewer mUfX: $6\rightarrow6$.
+ Reviewer axY1: $8\rightarrow8$.

---

### Decision · Program_Chairs · 2026-01-26

Accept (Oral)